# Discrete Neural Flow Samplers with Locally Equivariant Transformer

**Zijing Ou**[1], **Ruixiang Zhang**[2], **Yingzhen Li**[1]
[1]Imperial College London, [2]Apple
{z.ou22, yingzhen.li}@imperial.ac.uk  ruixiang_zhang2@apple.com

## Abstract

Sampling from unnormalised discrete distributions is a fundamental problem across various domains. While Markov chain Monte Carlo offers a principled approach, it often suffers from slow mixing and poor convergence. In this paper, we propose Discrete Neural Flow Samplers (DNFS), a trainable and efficient framework for discrete sampling. DNFS learns the rate matrix of a continuous-time Markov chain such that the resulting dynamics satisfy the Kolmogorov equation. As this objective involves the intractable partition function, we then employ control variates to reduce the variance of its Monte Carlo estimation, leading to a coordinate descent learning algorithm. To further facilitate computational efficiency, we propose locally equivaraint Transformer, a novel parameterisation of the rate matrix that significantly improves training efficiency while preserving powerful network expressiveness. Empirically, we demonstrate the efficacy of DNFS in a wide range of applications, including sampling from unnormalised distributions, training discrete energy-based models, and solving combinatorial optimisation problems.

## 1 Introduction

We consider the task of sampling from a discrete distribution $\pi(x) = \frac{\rho(x)}{Z}$, known only up to a normalising constant $Z = \sum_x \rho(x)$. This problem is fundamental in a wide range of scientific domains, including Bayesian inference (Murray et al., 2012), statistical physics (Newman & Barkema, 1999), and computational biology (Lartillot & Philippe, 2004). However, efficient sampling from such unnormalised distributions remains challenging, especially when the state space is large and combinatorially complex, making direct enumeration or exact computation of $Z$ infeasible.

Conventional sampling techniques, such as Markov Chain Monte Carlo (MCMC) (Metropolis et al., 1953) have been widely employed with great success. Nevertheless, MCMC often suffers from poor mixing and slow convergence due to the issues of Markov chains getting trapped in local minima and large autocorrelation (Neal et al., 2011). These limitations have motivated the development of neural samplers (Wu et al., 2020; Vargas et al., 2024; Máté & Fleuret, 2023), which leverage deep neural networks to improve sampling efficiency and convergence rates. In discrete settings, autoregressive models (Box et al., 2015) have been successfully applied to approximate Boltzmann distributions of spin systems in statistical physics (Wu et al., 2019). Inspired by recent advances in discrete diffusion models (Austin et al., 2021; Sun et al., 2023c; Campbell et al., 2022), Sanokowski et al. (2024, 2025) propose diffusion-based samplers with applications to solving combinatorial optimisation problems. Moreover, Holderrieth et al. (2025) introduces an alternative discrete sampler by learning a parametrised continuous-time Markov chain (CMCT) (Norris, 1998) to minimise the variance of importance weights between the CMCT-induced distribution and the target distribution.

Building on these advances, the goal of our paper is to develop a sampling method for discrete distributions that is both efficient and scalable. To this end, we introduce Discrete Neural Flow

Samplers (DNFS), a novel framework that learns the rate matrix of a CTMC whose dynamics satisfy the Kolmogorov forward equation (Oksendal, 2013). In contrast to discrete flow models (Campbell et al., 2024; Gat et al., 2024), which benefit from access to training data to fit the generative process, DNFS operates in settings where no data samples are available. This data-free setting makes direct optimisation of the Kolmogorov objective particularly challenging and necessitates new methodological advances to ensure stable and effective training. Specifically, the first difficulty lies in the dependence of the objective on the intractable partition function. We mitigate this by using control variates (Geffner & Domke, 2018) to reduce the variance of its Monte Carlo estimate, which enables efficient optimisation via coordinate descent. More critically, standard neural network parameterisations of the rate matrix render the objective computationally prohibitive. To make learning tractable, a locally equivariant Transformer architecture is introduced to enhance computational efficiency significantly while retaining strong model expressiveness. Empirically, DNFS proves to be an effective sampler for discrete unnormalised distributions. We further demonstrate its versatility in diverse applications, including training discrete energy-based models and solving combinatorial optimization problems.

## 2 Preliminaries

We begin by introducing the key preliminaries: the Continuous Time Markov Chain (CTMC) (Norris, 1998) and the Kolmogorov forward equation (Oksendal, 2013). Let $x$ be a sample in the $d$-dimensional discrete space $\{1, \ldots, S\}^d \triangleq \mathcal{X}$. A continuous-time discrete Markov chain at time $t$ is characterised by a rate matrix $R_t : \mathcal{X} \times \mathcal{X} \mapsto \mathbb{R}$, which captures the instantaneous rate of change of the transition probabilities. Specifically, the entries of $R_t$ are defined by

$$R_t(y, x) = \lim_{\Delta t \to 0} \frac{p_{t+\Delta t|t}(y|x) - \mathbf{1}_{y=x}}{\Delta t}, \quad \mathbf{1}_{y=x} = \begin{cases} 1, & y = x \\ 0, & y \neq x \end{cases}, \tag{1}$$

which equivalently yields the local expansion $p_{t+\Delta t|t}(y|x) = \mathbf{1}_{y=x} + R_t(y,x)\Delta t + o(t)$ and the rate matrix satisfies $R_t(y,x) \geq 0$ if $y \neq x$ and $R_t(x,x) = -\sum_{y \neq x} R_t(y,x)$. Given $R_t$, the marginal distribution $p_t(x_t)$ for any $t \in \mathbb{R}$ is uniquely determined. Let $x_{0 \leq t \leq 1}$ be a sample trajectory. Our goal is to seek a rate matrix $R_t$ that transports an initial distribution $p_0 \propto \eta$ to the target distribution $p_1 \propto \rho$. The trajectory then can be obtained via the Euler method (Sun et al., 2023c)

$$x_{t+\Delta t} \sim \mathrm{Cat}\left(x; \mathbf{1}_{x_{t+\Delta t}=x} + R_t(x_{t+\Delta t}, x)\Delta t\right), \quad x_0 \sim p_0, \tag{2}$$

and the induced probability path $p_t$ by $R_t$ satisfies the Kolmogorov equation (Oksendal, 2013)

$$\partial_t p_t(x) = \sum_y R_t(x,y)p_t(y) = \sum_{y \neq x} R_t(x,y)p_t(y) - R_t(y,x)p_t(x). \tag{3}$$

In this case, we say that the rate matrix $R_t$ generates the probability path $p_t$. Dividing both sides of the Kolmogorov equation by $p_t$ leads to

$$\partial_t \log p_t(x) = \sum_{y \neq x} R_t(x,y)\frac{p_t(y)}{p_t(x)} - R_t(y,x). \tag{4}$$

In the next section, we describe how to leverage Equation (4) to learn a model-based rate matrix for sampling from a given target distribution $\pi$, followed by the discussion of applications to discrete energy-based modelling and combinatorial optimisation.

## 3 Discrete Neural Flow Samplers

The rate matrix $R_t$ that transports the initial distribution to the target is generally not unique. However, we can select a particular path by adopting an annealing interpolation between the prior $\eta$ and the target $\rho$, defined as $p_t \propto \rho^t \eta^{1-t} \triangleq \tilde{p}_t$ (Gelman & Meng, 1998; Neal, 2001). This annealing path coincides with the target distribution $\pi \propto \rho$ at time $t = 1$. To construct an $R_t$ that generates the probability path $p_t$, we seek a rate matrix that satisfies the Kolmogorov equation in Equation (4). Specifically, we learn a model-based rate matrix $R_t^\theta(y, x)$, parametrised by $\theta$, by minimizing the loss

$$\mathcal{L}(\theta) = \mathbb{E}_{w(t), q_t(x)} \delta_t^2(x; R_t^\theta), \quad \delta_t(x; R_t^\theta) \triangleq \partial_t \log p_t(x) + \sum_{y \neq x} R_t^\theta(y,x) - R_t^\theta(x,y)\frac{p_t(y)}{p_t(x)}, \tag{5}$$

where $q_t$ is an arbitrary reference distribution that has the same support as $p_t$, and $w(t)$ denotes a time schedule distribution. At optimality, the condition $\delta_t(x; R_t^\theta) = 0$ holds for all $t$ and $x \in \mathcal{X}$, implying that the learned rate matrix $R_t^\theta$ ensures the dynamics prescribed by the Kolmogorov equation are satisfied along the entire interpolation path. In practice, minimising the loss (5) guides $R_t^\theta$ to correctly capture the infinitesimal evolution of the distribution $p_t$, enabling accurate sampling from the target distribution via controlled stochastic dynamics.

However, evaluating Equation (5) directly is computationally infeasible due to the intractable summation over $y$, which spans an exponentially large space of possible states, resulting in a complexity of $\mathcal{O}(S^d)$. To alleviate this issue, we follow Sun et al. (2023c); Campbell et al. (2022); Lou et al. (2024) by assuming independence across dimensions. In particular, we restrict the rate matrix $R_t^\theta$ such that it assigns non-zero values only to states $y$ that differ from $x$ in at most one dimension. Formally, $R_t^\theta(y, x) = 0$ if $y \notin \mathcal{N}(x)$, where $\mathcal{N}(x) := \{y \in \mathcal{X} | y_i \neq x_i \text{ at most one } i\}$. To improve clarity in the subsequent sections, we renote the rate matrix $R_t^\theta$ for $y \in \mathcal{N}(x)$ with $y_i \neq x_i$ as

$$R_t^\theta(y, x) \triangleq R_t^\theta(y_i, i|x) \quad \text{and} \quad R_t^\theta(x_i, i|x) = -\sum_{i, y_i \neq x_i} R_t^\theta(y_i, i|x), \tag{6}$$

which yields a simplified and more tractable form of the loss in Equation (5)

$$\delta_t(x; v_t) = \partial_t \log p_t(x) + \sum_{i, y_i \neq x_i} R_t^\theta(y_i, i|x) - R_t^\theta(x_i, i|y) \frac{p_t(y)}{p_t(x)}. \tag{7}$$

This approximation reduces the computational complexity from $\mathcal{O}(S^d)$ to $\mathcal{O}(S \times d)$. Nonetheless, two main challenges persist. First, the time derivative $\partial_t \log p_t(x)$ remains intractable due to the dependence on the partition function, as it expands to $\partial_t \log \tilde{p}_t(x) - \partial_t \log Z_t$ with $Z_t = \sum_x \tilde{p}_t(x)$ being intractable. Second, evaluating Equation (7) requires evaluating the neural network $|\mathcal{N}(x)|$ times, which is computationally expensive for each $x$. In the following, we propose several techniques to address these computational bottlenecks.

## 3.1 Estimating the Time Derivative of the Log-Partition Function

To estimate the time derivative, note that $\partial_t \log Z_t = \mathbb{E}_{p_t(x)}[\partial_t \log \tilde{p}_t(x)]$, which can be approximated via the Monte Carlo estimator $\partial_t \log Z_t \approx \frac{1}{K} \sum_{k=1}^{K} \partial_t \log \tilde{p}_t(x_t^{(k)})$. However, this approach relies on sampling from $p_t$, which is typically impractical due to the lack of convergence guarantees for short-run MCMC in practice and the high variance inherent in Monte Carlo estimation. To address this issue, we leverage a key identity that holds for any given rate matrix $R_t$

$$\partial_t \log Z_t = \underset{c_t}{\text{argmin}} \, \mathbb{E}_{p_t}(\xi_t(x; R_t) - c_t)^2, \xi_t(x; R_t) \triangleq \partial_t \log \tilde{p}_t(x_t) - \sum_y R_t(x, y) \frac{p_t(y)}{p_t(x)} \tag{8}$$

Because the objective (8) is convex in $c_t$, the minimizer is given by $\partial_t \log Z_t = \mathbb{E}_{p_t} \xi_t(x; R_t)$. Moreover, in practice, the expectation over $p_t$ can be safely replaced by an expectation over any distribution $q_t$ with the same support, still yielding a valid estimate of the time derivative (see Appendix A.1 for details). Empirically, we observe that using Equation (8) results in a significantly lower-variance estimator compared to the direct Monte Carlo approach

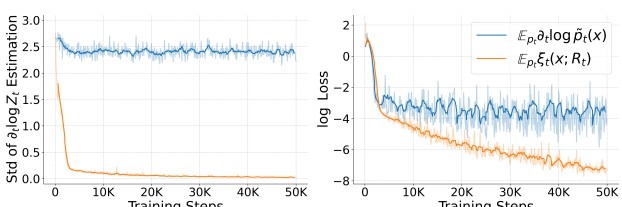

Figure 1: Comparison of std. dev. and training loss for different estimators of $\partial_t \log Z_t$. Lower variance estimator exhibits lower training loss, indicating a better learned rate matrix satisfying the Kolmogorov equation in Equation (5).

$\partial_t \log Z_t = \mathbb{E}_{p_t(x)}[\partial_t \log \tilde{p}_t(x)]$. This reduction in variance can lead to improved optimisation performance. To assess this, we conducted experiments on the Ising model (Mézard et al., 1987), minimising the loss in Equation (7) using two different estimators for $\partial_t \log Z_t$. The standard deviations of both estimators, as well as their corresponding loss values during training, are plotted over training steps in Figure 1. The results demonstrate that the estimator based on $\mathbb{E}_{p_t} \xi_t(x; R_t)$ consistently achieves lower loss values, underscoring the benefits of reduced variance in estimating $\partial_t \log Z_t$ for

improved training dynamics. In Appendix A.2, we provide a perspective of control variate (Geffner & Domke, 2018) to further explain this observation. This insight enables a coordinate descent approach to learning the rate matrix. Specifically: i) $\theta \leftarrow \operatorname{argmin}_\theta \int_0^1 \mathbb{E}_{q_t(x)} (\xi_t(x; R_t^\theta) - c_t)^2 \, dt$; and ii) $c_t \leftarrow \operatorname{argmin}_{c_t} \mathbb{E}_{q_t}(\xi_t(x; R_t) - c_t)^2$. Alternatively, the time derivative can be parameterised directly via a neural network $c_t^\phi$, allowing joint optimisation of $\theta$ and $\phi$ through the objective $\operatorname{argmin}_{\theta,\phi} = \int_0^1 \mathbb{E}_{q_t(x)}(\xi_t(x; R_t^\theta) - c_t^\phi)^2 \, dt$. This formulation recovers the physics-informed neural network (PINN) loss proposed in Holderrieth et al. (2025). A detailed discussion of the connection to the PINN loss is provided in Appendix A.3.

## 3.2 Efficient Training with Locally Equivariant Networks

As previously noted, computing the $\delta$ function in Equation (7) requires evaluating the neural network $|\mathcal{N}|$ times, which is computationally prohibitive. Inspired by Holderrieth et al. (2025), we proposed to mitigate this issue by utilising locally equivariant networks, an architectural innovation that significantly reduces the computational complexity with the potential to preserve the capacity of network expressiveness. A central insight enabling this reduction is that any rate matrix can be equivalently expressed as a one-way rate matrix[1], while still inducing the same probabilistic path. This is formalised in the following proposition:

**Proposition 1.** *For a rate matrix $R_t$ that generates the probabilistic path $p_t$, there exists a one-way rate matrix $Q_t(y, x) = \left[ R_t(y, x) - R_t(x, y) \frac{p_t(y)}{p_t(x)} \right]_+$ if $y \neq x$ and $Q_t(x, x) = \sum_{y \neq x} Q_t(y, x)$, that generates the same probabilistic path $p_t$, where $[z]_+ = \max(z, 0)$ denotes the ReLU operation.*

This result was originally introduced by Zhang et al. (2023b), and we include a proof in Appendix B.1 for completeness. Building on Proposition 1, we can parameterise $R_t^\theta$ directly as a one-way rate matrix. To achieve this, we use a locally equivariant neural network as described by Holderrieth et al. (2025). Specifically, a neural network $G$ is locally equivariant if and only if:

$$G_t^\theta(\tau, i|x) = -G_t^\theta(x_i, i|\text{Swap}(x, i, \tau)), \quad i = 1, \ldots, d \tag{9}$$

where $\text{Swap}(x, i, \tau) = (x_1, \ldots, x_{i-1}, \tau, x_{i+1}, \ldots, x_d)$ and $\tau \in \{1, \ldots, S\} \triangleq \mathcal{S}$. Based on this, the one-way rate matrix can be defined as $R_t^\theta(\tau, i|x) \triangleq [G_t^\theta(\tau, i|x)]_+$. Substituting this parametrization into Equation (7), we obtain the simplified expression:

$$\delta_t(x; R_t^\theta) = \partial_t \log p_t(x) + \sum_{i, y_i \neq x_i} [G_t^\theta(y_i, i|x)]_+ - [-G_t^\theta(y_i, i|x)]_+ \frac{p_t(y)}{p_t(x)}. \tag{10}$$

This formulation reduces the computational cost from $\mathcal{O}(|\mathcal{N}|)$ to $\mathcal{O}(1)$, enabling far more efficient training. We term the proposed method as discrete neural flow sampler (DNFS) and summarise the training and sampling details in Appendix C. Nonetheless, the gain in efficiency introduces challenges in constructing a locally equivariant network (leNet) that is both expressive and flexible.

## 3.3 Instantiation of leNets: Locally Equivariant Transformer

To construct a locally equivariant network, we first introduce *hollow network* (Chen & Duvenaud, 2019). Formally, let $x_{i \leftarrow \tau} = (x_1, \ldots, x_i = \tau, \ldots, x_d)$ denote the input with its $i$-th token set to $\tau$. A function $H : \mathcal{X} \mapsto \mathbb{R}^{d \times h}$ is termed a hollow network if it satisfies $H(x_{i \leftarrow \tau})_{i,:} = H(x_{i \leftarrow \tau'})_{i,:}, \forall \tau, \tau' \in \mathcal{S}$, where $M_{i,:}$ denotes the $i$-th row of the matrix $M$. Intuitively, it implies that the output at position $i$ is invariant to the value of the $i$-th input token. Hollow networks provide a foundational building block for constructing locally equivariant networks, as formalised in the following proposition.

**Proposition 2** (Instantiation of Locally Equivariant Networks)**.** *Let $x \in \mathcal{X}$ denote the input tokens and $H : \mathcal{X} \mapsto \mathbb{R}^{d \times h}$ be a hollow network. Furthermore, for each token $\tau \in \mathcal{S}$, let $\omega_\tau \in \mathbb{R}^h$ be a learnable projection vector. Then, the locally equivariant network can be constructed as:*

$$G(\tau, i|x) = (\omega_\tau - \omega_{x_i})^T H(x)_{i,:}.$$

---

[1]A rate matrix $R$ is said to be one-way if $R(y, x) > 0$ implies $R(x, y) = 0$. That is, if a transition from $x$ to $y$ is permitted, the reverse transition must be impossible.

This can be verified via $G(\tau, i|x) = -(\omega_{x_i} - \omega_\tau)^T H(\text{Swap}(x, i, \tau))_{i,:} = -G(x_i, i|\text{Swap}(x, i, \tau))$. Although Proposition 2 offers a concrete approach to constructing locally equivariant networks, significant challenges persist. In contrast to globally equivariant architectures (Cohen & Welling, 2016; Fuchs et al., 2020), where the composition of equivariant layers inherently preserves equivariance, locally equivariant networks are more delicate to design. In particular, stacking locally equivariant layers does not, in general, preserve local equivariance. While Holderrieth et al. (2025) propose leveraging multi-layer perceptions (MLPs), attention mechanisms, and convolutional layers (see Appendix B.2 for details) to construct locally equivariant networks, these architectures may still fall short in terms of representational capacity and flexibility.

**Locally Equivariant Transformer.**
As shown in Proposition 2, a crucial component to construct a locally equivariant network is the hollow network. Specifically, the key design constraint is that the network's output at dimension $i$ must be independent of the corresponding input value $x_i$. Otherwise, any dependence would result in information leakage and violate local equivariance. However, the $i$-th output may depend freely on all other coordinates of the input token $x$ except for the $i$-th entry. This insight motivates the use of hollow transformers (Sun et al., 2023c) as a foundation for constructing locally equivariant networks. Specifically, it employs

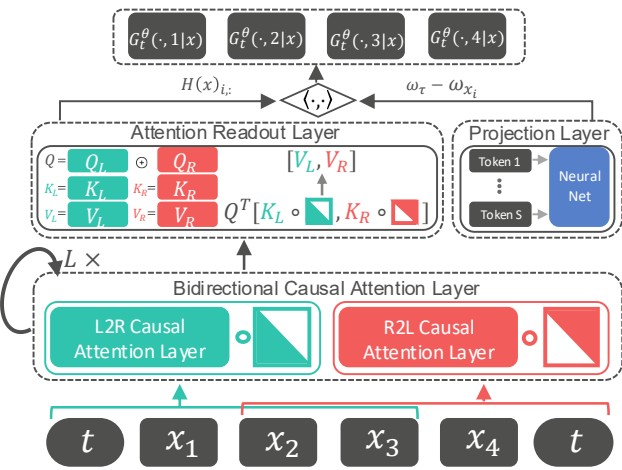

Figure 2: Illustration of the leTF network.

two autoregressive Transformers (Vaswani et al., 2017; Radford et al., 2018) per layer; one processing inputs from left to right, and the other from right to left. In the readout layer, the representations from two directions are fused via attention to produce the output. This design ensures that each output dimension remains independent of its corresponding input coordinate, while still leveraging the expressiveness of multi-layer Transformers. Thereby, the final output $G_t^\theta(\cdot, i|x)$ can be obtained by taking the inner product between the hollow attention output and the token embeddings produced by the projection layer. We term the proposed architecture as locally equivariant transformer (leTF), and defer the implementation details to Appendix B.3.

**Comparison of Different leNets.** In Figure 3, we compare leTF with other locally equivariant networks for training a discrete neural flow sampler on the Ising model. The results show that leTF achieves lower estimation errors and higher effective sample sizes (see Appendix E.1 for experimental details). leAttn and leMLP, which each consist of only a single locally equivariant layer, perform sig-

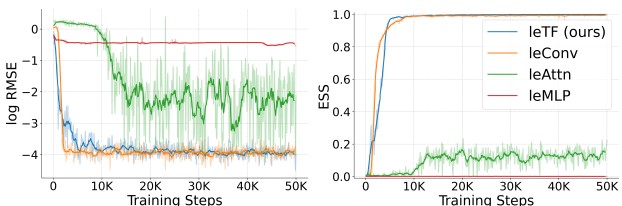

Figure 3: Comparison of log RMSE ($\downarrow$) and ESS ($\uparrow$) for different locally equivariant networks. More expressive networks achieve better performance.

nificantly worse, highlighting the importance of network expressiveness in achieving effective local equivariance. Although leConv performs comparably to leTF, its convolutional design is inherently less flexible. It is restricted to grid-structured data, such as images or the Ising model, and does not readily generalise to other data types like text or graphs. Additionally, as shown in Figure 14, leTF achieves lower training loss compared to leConv, further confirming its advantage in expressiveness.

# 4 Applications and Experiments

To support our theoretical discussion, we first evaluate the proposed methods by sampling from predefined unnormalised distributions. We then demonstrate two important applications to DNFS: i)

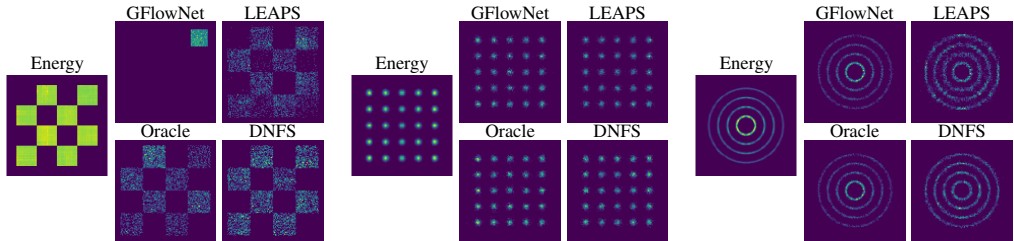

Figure 4: Comparison between different discrete samplers on pre-trained EBMs.

training discrete energy-based models and ii) solving combinatorial optimisation problems. Detailed experimental settings and additional results are provided in Appendix E.

## 4.1 Sampling from Unnormalised Distributions

**Sampling from Pre-trained EBMs.** We begin by evaluating the effectiveness of our method by sampling from a pre-trained deep energy-based model. Specifically, we train an EBM on 32-dimensional binary data obtained by applying the Gray code transformation (Waggener & Waggener, 1995) to a 2D continuous plane, following Dai et al. (2020). The EBM consists of a 4-layer MLP with 256 hidden units per layer and is trained using the discrete energy discrepancy introduced in Schröder et al. (2024). Therefore, the trained EBM defines an unnormalised distribution, upon which we train a discrete neural sampler. We benchmark DNFS against three baselines: (i) long-run Gibbs sampling (Casella & George, 1992) as the oracle; (ii) GFlowNet with trajectory balance (Malkin et al., 2022); and (iii) LEAPS (Holderrieth et al., 2025) with the proposed leTF network.

The results, shown in Figure 4, demonstrate that the proposed method, DNFS, produces samples that closely resemble those from the oracle Gibbs sampler. In contrast, GFlowNet occasionally suffers from mode collapse, particularly on structured datasets such as the checkerboard pattern. Although LEAPS with leTF achieves performance comparable to DNFS, it sometimes produces inaccurate samples that fall in smoother regions of the energy landscape, potentially due to imprecise estimation of $\partial_t \log Z_t$. Furthermore, we observe that LEAPS with leConv performs poorly in this setting (see Figure 12), reinforcing the limited expressiveness of locally equivariant convolutional networks when applied to non-grid data structures. For a more comprehensive evaluation, additional visualisations on other datasets are provided in Figure 11, further illustrating the effectiveness of our method.

**Sampling from Ising Models.** We further evaluate our method on the task of sampling from the lattice Ising model, which has the form of

$$p(x) \propto \exp(x^T J x), x \in \{-1, 1\}^D, \tag{11}$$

where $J = \sigma A_D$ with $\sigma \in \mathbb{R}$ and $A_D$ being the adjacency matrix of a $D \times D$ grid.[2] In Figure 5, we evaluate DNFS on a $D = 10 \times 10$ lattice grid with $\sigma = 0.1$, comparing it to baselines methods in terms of effective sample size (ESS) (see Appendix D.1 for details) and the energy histogram of $5,000$ samples. The oracle energy distribution is approximated using long-run Gibbs sampling. The results show that DNFS performs competitively with LEAPS and significantly outperforms GFlowNet, which fails to capture the correct mode of the energy distribution. Although LEAPS with leConv achieves a comparable effective sample size, it yields a less accurate approximation of the energy distribution compared to DNFS. Furthermore, DNFS attains a lower loss value, as shown in Figure 15.

These findings underscore the effectiveness of the proposed neural sampler learning algorithm and highlight the strong generalisation capability of leTF across both grid-structured and non-grid data. For a more comprehensive evaluation, we also compare our method with MCMC-based approaches in Figure 13, further demonstrating the efficacy of our approach.

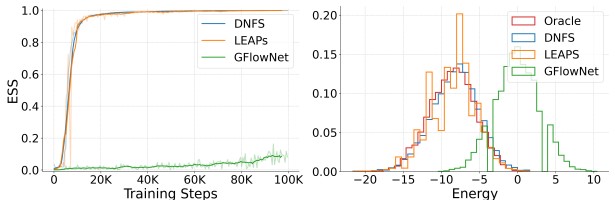

Figure 5: Comparison of effective sample size and histogram of sample energy on the lattice Ising model.

[2]The adjacency matrix is constructed using A_D = igraph.Graph.Lattice(dim=[D, D], circular=True).

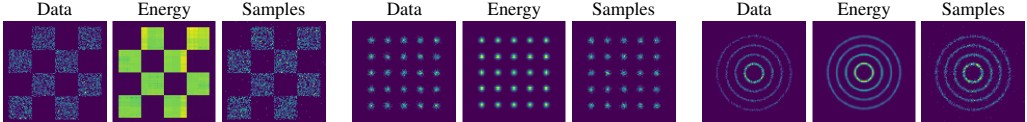

Figure 6: Results of probability mass estimation in training discrete EBMs. We visualise the training data, learned energy landscape, and the synthesised samples of DNFS.

## 4.2 Training Discrete Energy-based Models

A key application of DNFS is training energy-based models (EBMs). Specifically, EBMs define a parametric distribution $p_\phi \propto \exp(-E_\phi(x))$, where the goal is to learn an energy function $E_\phi$ that approximates the data distribution. EBMs are typically trained using contrastive divergence (Hinton, 2002), which estimates the gradient of the log-likelihood as

$$\nabla_\phi \log p_\phi(x) = \mathbb{E}_{p_\phi(y)}[\nabla_\phi E_\phi(y)] - \nabla_\phi E_\phi(x). \tag{12}$$

To approximate this intractable gradient, we train a rate matrix $R_t^\theta$ to sample from the target $p_\phi$. This enables using importance sampling to estimate the expectation (see Appendix D.1 for details).

$$\mathbb{E}_{p_\phi(x)}[\nabla_\phi E_\phi(x)] \approx \sum_{k=1}^{K} \frac{\exp(w^{(k)})}{\sum_{j=1}^{K} \exp(w^{(j)})} \nabla_\phi E_\phi(x^{(k)}), \quad w^{(k)} = \int_0^1 \xi_t(x_t; R_t^\theta) \, \mathrm{d}t. \tag{13}$$

This neural-sampler-based approach is more effective than traditional MCMC methods, as in optimal training, it has garuantee to produce exact samples from the target distribution within a fixed number of sampling steps. Moreover, neural samplers are arguabelly easier to discover regularities and jump between modes compared to MCMC methods, leading to better exploration of the whole energy landscape, and thus results in a more accurate estimate of the energy function (Zhang et al., 2022a). To demonstrate the effectiveness of DNFS in energy-based modelling, we conduct experiments on probability mass estimation with synthetic data and training Ising models.

**Probability Mass Estimation.** Following Dai et al. (2020), we first generate 2D floating-point data from several two-dimensional distributions. Each dimension is then encoded using a 16-bit Gray code, resulting in a 32-dimensional training dataset with 2 possible states per dimension.

Figure 6 illustrates the estimated energy landscape alongside samples generated using the trained DNFS sampler. The results demonstrate that the learned EBM effectively captures the multi-modal structure of the underlying distribution, accurately modelling the energy across the data support. The sampler produces samples that closely resemble the training data, highlighting the effectiveness of DNFS in training discrete EBMs. Additional qualitative results are presented in Figure 16. In Table 3, we provide a quantitative comparison of our method with several baselines, focusing in particular on two contrastive divergence (CD)-based approaches: PCD (Tieleman, 2008) with MCMC and ED-GFN (Zhang et al., 2022a) with GFlowNet. Our method, built upon the proposed DNFS, consistently outperforms PCD in most settings, underscoring the effectiveness of DNFS in training energy-based models. While ED-GFN achieves better performance than DNFS, it benefits from incorporating a Metropolis-Hastings (MH) (Hastings, 1970) correction to sample from the model distribution $p_\phi$ in Equation (12), which may offer an advantage over the importance sampling strategy used in Equation (13). We leave the integration of MH into DNFS as a direction for future work.

**Training Ising Models.** We further assess DNFS for training the lattice model defined in Equation (11). Following Grathwohl et al. (2021), we generate training data using Gibbs sampling and use these samples to learn a symmetric matrix $J_\phi$ to estimate the true matrix in the Ising model. Importantly, the training algorithms do not have access to the true data-generating matrix $J$, but only to the synthesised samples.



Figure 7: Results on learning Ising models.

In Figure 7, we consider a $D = 10 \times 10$ grid with $\sigma = 0.1$ and visualise the learned matrix $J_\phi$ using a heatmap. The results show that the proposed method successfully captures the underlying pattern of the ground truth, demonstrating the effectiveness of DNFS. Further quantitative analysis across various configurations of $D$ and $\sigma$ is presented in Table 5.

Table 1: Max independent set experimental results. We report the absolute performance, approximation ratio (relative to GUROBI), and inference time.

| METHOD | ER16-20 | | | ER32-40 | | | ER64-75 | | |
|---|---|---|---|---|---|---|---|---|---|
| | SIZE ↑ | DROP ↓ | TIME ↓ | SIZE ↑ | DROP ↓ | TIME ↓ | SIZE ↑ | DROP ↓ | TIME ↓ |
| GUROBI | 8.92 | 0.00% | 4:00 | 14.62 | 0.00% | 4:03 | 20.55 | 0.00% | 4:10 |
| RANDOM | 5.21 | 41.6% | 0:03 | 6.31 | 56.8% | 0:06 | 8.63 | 58.0% | 0:09 |
| DMALA | 8.81 | 1.23% | 0:21 | 14.02 | 4.10% | 0:22 | 19.54 | 4.91% | 0:24 |
| GFLOWNET | 8.75 | 1.91% | 0:02 | 13.93 | 4.72% | 0:04 | 19.13 | 6.91% | 0:07 |
| DNFS | 8.28 | 7.17% | 0:03 | 13.18 | 9.85% | 0:06 | 18.12 | 11.8% | 0:09 |
| DNFS+DMALA | 8.91 | 0.11% | 0:10 | 14.31 | 2.12% | 0:15 | 20.06 | 2.38% | 0:22 |

## 4.3 Solving Combinatorial Optimisation Problems

Another application of DNFS is solving combinatorial optimisation problems. As an example, we describe how to formulate the maximum independent set as a sampling problem.

**Maximum Independent Set as Sampling.** Given a graph $G = (V, E)$, the maximum independent set (MIS) problem aims to find the largest subset of non-adjacent vertices. It can be encoded as a binary vector $x \in \{0, 1\}^{|V|}$, where $x_i = 1$ if vertex $i$ is included in the set, and $x_i = 0$ otherwise. The objective is to maximise $\sum_{i=1}^{|V|} x_i$ subject to $x_i x_j = 0$ for all $(i, j) \in E$. This can be formulated as sampling from the following unnormalised distribution:

$$p(x) \propto \exp\left(\frac{1}{T}\left(\sum_{i=1}^{|V|} x_i - \lambda \sum_{(i,j)\in E} x_i x_j\right)\right), \tag{14}$$

where $T > 0$ is the temperature and $\lambda > 1$ is a penalty parameter enforcing the independence constraint. As $T \to 0$, $p(x)$ uniformly concentrates on the maximum independent sets.

Therefore, we can train DNFS to sample from $p(x)$, which will produce high-quality solutions to the MIS problem. To enable generalisation across different graphs $G$, we condition the locally equivariant transformer on the graph structure. Specifically, we incorporate the Graphformer architecture (Ying et al., 2021), which adjusts attention weights based on the input graph. This allows the model to adapt to varying graph topologies. We refer to this architecture as the locally equivariant Graphformer (leGF), with implementation details provided in Appendix D.4.

**Experimental Settings.** In this experiment, we apply our method to solve the Maximum Independent Set (MIS) problem, with other settings deferred to Appendix E.3. Specifically, we benchmark MIS on Erdős–Rényi (ER) random graphs (Erdos, 1961), comprising 1,000 training and 100 testing instances, each with 16 to 75 vertices. Evaluation on the test set includes both performance and inference time. We report the average solution size and the approximation ratio with respect to the best-performing mixed-integer programming solver (GUROBI) (Gurobi, 2023), which serves as the oracle.

**Results & analysis.** We compare our method against two baselines: an annealed MCMC sampler (Sun et al., 2023b) using DMALA (Zhang et al., 2022b), and a neural sampler based on GFlowNet (Zhang et al., 2023a). Additionally, we include results from a randomly initialised version of DNFS without training, which serves as an estimate of the task's intrinsic difficulty. As shown in Table 1, DNFS after training substantially outperforms its untrained counterpart, highlighting the effectiveness of our approach. While the MCMC-based method achieves the strongest overall performance, it requires longer inference time. Compared to GFlowNet, another neural sampler, DNFS performs slightly worse. This may be attributed to the fact that GFlowNet restricts sampling to feasible solutions only along the trajectory, effectively reducing the exploration space and making the learning problem easier. Incorporating this inductive bias into DNFS is a promising direction for future work. Nevertheless, a key advantage of our method is that the unnormalised marginal distribution $p_t$ is known, allowing us to integrate additional MCMC steps to refine the sampling trajectory. As shown in the last row of Table 1, this enhancement leads to a substantial performance gain. Further analysis of this approach is provided in Table 7 in the appendix.

# 5 Related Work

**CTMCs and Discrete Diffusion.** Our work builds on the framework of continuous-time Markov chains (CTMCs), which were first introduced in generative modelling by Austin et al. (2021); Sun et al. (2023c); Campbell et al. (2022) under the context of continuous-time discrete diffusion models, where the rate matrix is learned from training data. This approach was later simplified and generalised to discrete-time masked diffusion (Shi et al., 2024; Sahoo et al., 2024; Ou et al., 2024), demonstrating strong performance across a wide range of applications, including language modelling (Lou et al., 2024; Zhang et al., 2025), molecular simulation (Campbell et al., 2024), and code generation (Gat et al., 2024; Gong et al., 2025). However, these methods require training data and are inapplicable when only an unnormalised target is given.

**MCMC and Neural Samplers.** Markov chain Monte Carlo (MCMC) (Metropolis et al., 1953) is the *de facto* approach to sampling from a target distribution. In discrete spaces, Gibbs sampling (Casella & George, 1992) is a widely adopted method. Building on this foundation, Zanella (2020) improve the standard Gibbs method by incorporating locally informed proposals to improve sampling efficiency. This method was extended to include gradient information to drastically reduce the computational complexity of flipping bits in several places. This idea was further extended by leveraging gradient information (Grathwohl et al., 2021; Sun et al., 2022a), significantly reducing the computational cost. Inspired by these developments, discrete analogues of Langevin dynamics have also been introduced to enable more effective sampling in high-dimensional discrete spaces (Zhang et al., 2022b; Sun et al., 2023a). Despite their theoretical appeal, MCMC methods often suffer from slow mixing and poor convergence in practice. To address these limitations, recent work has proposed neural samplers, including diffusion-based (Vargas et al., 2024; Chen et al., 2024; Richter & Berner, 2024) and flow-based (Máté & Fleuret, 2023; Tian et al., 2024; Chen et al., 2025) approaches. However, the majority of these methods are designed for continuous spaces, and there remains a notable gap in the literature when it comes to sampling methods for discrete distributions. A few exceptions include Sanokowski et al. (2024, 2025), which are inspired by discrete diffusion models and primarily target combinatorial optimisation problems. A concurrent work, MDNS (Zhu et al., 2025), introduces a masked diffusion neural sampler grounded in stochastic optimal control theory (Berner et al., 2022). LEAPS (Holderrieth et al., 2025) and our method DNFS are more closely related to discrete flow models (Campbell et al., 2024; Gat et al., 2024), as both can be view as learning a CTMC to satisfy the Kolmogorov forward equation. While LEAPS parametrise $\partial_t \log Z_t$ using a neural network, DNFS estimates it via coordinate descent.

**Discrete EBMs and Neural Combinatorial Optimisation.** Contrastive divergence is the *de facto* approach to train energy-based models, but it relies on sufficiently fast mixing of Markov chains, which typically cannot be achieved (Nijkamp et al., 2020). To address this, several sampling-free alternatives have been proposed, including energy discrepancy (Schröder et al., 2023; Schröder et al., 2024), ratio matching (Lyu, 2012), and variational approaches (Lázaro-Gredilla et al., 2021). More recently, Zhang et al. (2022a) replace MCMC with GFlowNet, a neural sampler that arguably offers improvement by reducing the risk of getting trapped in local modes. Our work follows this line of research by using DNFS as a neural alternative to MCMC for training energy-based models. Sampling methods are also widely used to solve combinatorial optimisation problems (COPs). Early work (Sun et al., 2022b) demonstrated the effectiveness of MCMC techniques for this purpose. More recent approaches (Zhang et al., 2023a; Sanokowski et al., 2024, 2025) leverage neural samplers to learn amortised solvers for COPs. In this paper, we further show that DNFS is well-suited for combinatorial optimisation tasks, demonstrating its flexibility and broad applicability.

# 6 Conclusion and Limitation

We proposed discrete neural flow samplers (DNFS), a discrete sampler that learns a continuous-time Markov chain to satisfy the Kolmogorov forward equation. While our empirical studies demonstrate the effectiveness of DNFS across various applications, it also presents several limitations. A natural direction for future work is to extend DNFS beyond binary settings. However, this poses significant challenges due to the high computational cost of evaluating the ratios in Equation (7). As demonstrated in Appendix C.2, a naive Taylor approximation introduces bias into the objective, resulting in suboptimal solutions. Overcoming this limitation will require more advanced and principled approximation techniques. Additionally, we find that the current framework struggles to

scale to very high-dimensional distributions. This difficulty arises mainly from the summation over the ratios in Equation (7), which can lead to exploding loss values. Designing methods to stabilise this computation represents a promising avenue for future research. Finally, extending DNFS to the masked diffusion setting offers another compelling direction, with the potential to support more flexible and efficient sampling over structured discrete spaces.

**Broader impact.** This paper aims to advance machine learning research. While there may be potential societal impacts, none require specific mention at this time.

## Acknowledgements

ZO is supported by the Lee Family Scholarship. We would like to thank Tobias Schröder and Sangwoong Yoon for their valuable discussions on the early draft of this work. ZO also thanks Peter Holderrieth for generously sharing his insights on LEAPS during their discussion at ICLR 2025.

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

# Appendix for "Discrete Neural Flow Samplers with Locally Equivariant Transformer"

## Contents

## A  Variance Reduction and Control Variates

In this section, we analyse the variance reduction phenomenon shown in Figure 1 through the lens of control variates. We begin with the proof of Equation (8) and the discussion of control variates, then establish its connection to LEAPS (Holderrieth et al., 2025).

### A.1  Proof of Equation (8)

Recall from Equation (8) that the following equation holds:

$$\partial_t \log Z_t = \operatorname*{argmin}_{c_t} \mathbb{E}_{p_t}(\xi_t(x; R_t) - c_t)^2, \xi_t(x; R_t) \triangleq \partial_t \log \tilde{p}_t(x_t) - \sum_y R_t(x, y)\frac{p_t(y)}{p_t(x)}.$$

We now provide a detailed proof of this result, beginning with two supporting lemmas.

**Lemma 1** (Discrete Stein Identity). *Given a rate matrix $R(y, x)$ that satisfies $R(x, x) = -\sum_{y \neq x} R(y, x)$, we have the identity $\mathbb{E}_{p(x)} \sum_y R(x, y)\frac{p(y)}{p(x)} = 0$.*

*Proof.* To prove the result, notice that

$$\mathbb{E}_{p(x)}\sum_y R(x,y)\frac{p(y)}{p(x)} = \sum_x\sum_y R(x,y)p(y) = \sum_y p(y)\sum_x R(x,y) = 0,$$

which completes the proof. For a more comprehensive overview of the discrete Stein operator, see Shi et al. (2022). □

**Lemma 2.** *Let $c_t^* = \arg\min_{c_t} \mathbb{E}_{p_t}(\xi_t(x; R_t) - c_t)^2$, then $c_t^* = \mathbb{E}_{p_t}\xi_t(x; R_t)$.*

*Proof.* To see this, we can expand the objective

$$\mathcal{L}(c_t) = \arg\min_{c_t} \mathbb{E}_{p_t}(\xi_t(x; R_t) - c_t)^2 = c_t^2 - 2c_t\mathbb{E}_{p_t}\xi_t(x; R_t) + c = (c_t - \mathbb{E}_{p_t}\xi_t(x; R_t))^2 + c',$$

where the final expression is minimized when $c_t^* = \mathbb{E}_{p_t}\xi_t(x; R_t)$. □

We are now ready to prove Equation (8). Specifically:

$$\begin{aligned}
c_t^* &= \arg\min_{c_t}\mathbb{E}_{p_t}(\xi_t(x; R_t) - c_t)^2 \\
&= \mathbb{E}_{p_t}\xi_t(x; R_t) \\
&= \mathbb{E}_{p_t}\partial_t\log\tilde{p}_t(x) - \mathbb{E}_{p_t}\sum_y R_t(x,y)\frac{p_t(y)}{p_t(x)} \\
&= \mathbb{E}_{p_t}\partial_t\log\tilde{p}_t(x) = \partial_t\log Z_t,
\end{aligned}$$

where the second and third equations follow Lemmas 1 and 2 respectively.

### A.2 Discrete Stein Control Variates

To better understand the role of control variates (Geffner & Domke, 2018) in variance reduction as shown in Figure 1, let us consider a standard Monte Carlo estimation problem. Suppose our goal is to estimate the expectation $\mu = \mathbb{E}_\pi[f(x)]$, where $f(x)$ is a function of interest. A basic estimator for $\mu$ is the the Monte Carlo average $\hat{\mu} = \frac{1}{K}\sum_{k=1}^K f(x^{(k)}), x^{(k)} \sim \pi$. Now, suppose we have access to another function $g(x)$, known as a control variate, which has a known expected value $\gamma = \mathbb{E}_\pi[g(x)]$. We can the use $g(x)$ to construct a new estimator: $\check{\mu} = \frac{1}{K}\sum_{k=1}^K (f(x^{(k)}) - \beta g(x^{(k)})) + \beta\gamma$. This new estimator $\check{\mu}$ is *unbiased* for any choice of $\beta$, since $\mathbb{E}[\check{\mu}] = \mathbb{E}[f(x)] - \beta\mathbb{E}[g(x)] + \beta\gamma = \mu$. The benefit of this construction lies not in bias correction but in variance reduction. To see this, we can compute the variance of $\check{\mu}$:

$$\mathbb{V}[\check{\mu}] = \frac{1}{K}(\mathbb{V}[f] - 2\beta\mathrm{Cov}(f,g) + \beta^2\mathbb{V}[g]). \tag{15}$$

This is a quadratic function of $\beta$, and since it is convex, ts minimum can be found by differentiating w.r.t. $\beta$ and setting the derivative to zero. This yields the optimal coefficient $\beta^* = \mathrm{Cov}(f,g)/\mathbb{V}[g]$. Substituting it back into the variance expression (15) gives:

$$\mathbb{V}[\check{\mu}] = \frac{1}{K}\mathbb{V}[\hat{\mu}](1 - \mathrm{Corr}(f,g)^2). \tag{16}$$

This result shows a key insight: the effectiveness of a control variate depends entirely on its correlation with the target function $f$. As long as $f$ and $g$ are correlated (positively or negatively), the variance of $\check{\mu}$ is strictly less than that of $\hat{\mu}$. The stronger the correlation, the greater the reduction. In practice, the optimal coefficient $\beta'$ can be estimated from the same sample used to compute the Monte Carlo estimate, typically with minimal additional cost (Ranganath et al., 2014). However, the main challenge lies in selecting or designing a suitable control variate $g$ that both correlates well with $f$ and has a tractable expectation under $\pi$. For an in-depth treatment of this topic and practical considerations, see Geffner & Domke (2018).

Fortunately, Lemma 1 provides a principled way to construct a control variate tailored to our setting $\mathbb{E}_{p_t}[f(x)] \triangleq \mathbb{E}_{p_t}[\partial_t\log\tilde{p}_t(x)] \approx \frac{1}{K}\sum_{k=1}^K \partial_t\log\tilde{p}_t(x^{(k)})$, where $x^{(k)} \sim p_t$. To reduce the variance of this estimator, we seek a control variate $g(x)$ whose expectation under $p_t$ is known. Inspired by

the discrete Stein identity, we define $g(x) = \sum_y R_t(x,y)\frac{p_t(y)}{p_t(x)}$, which satisfies $\mathbb{E}_{p_t}[g(x)] = 0$ by Lemma 1. This makes $g$ a valid control variate with known mean. Using this construction, we can define a variance-reduced estimator as:

$$\check{\mu} = \frac{1}{K}\sum_{k=1}^{K}\partial_t\log\tilde{p}_t(x^{(k)}) - \beta^*\left(\sum_y R_t(x^{(k)},y)\frac{p_t(y)}{p_t(x^{(k)})}\right), \quad x^{(k)} \sim p_t. \tag{17}$$

This estimator remains unbiased for any $\beta$, but with the optimal choice $\beta^*$, it can substantially reduce variance. Moreover, in the special case where the parameter $\theta$ is optimal (in the sense that the objective in Equation (5) equals zero), an even stronger result emerges: the control variate becomes perfectly (negatively) correlated with the target function. That is, $g(x) = -f(x) + c$, where $c$ is a constant independent of the sample $x$, leading to $\text{Corr}(f,g) = -1$. In this idealised case, $\check{\mu}$ becomes a zero-variance estimator, a rare but highly desirable scenario. For a more comprehensive discussion of discrete Stein-based control variates and their applications in variance reduction, we refer the reader to Shi et al. (2022).

## A.3 Connection to LEAPS

Equation (8) provides a natural foundation for learning the rate matrix $R_t^\theta$ using coordinate ascent. This involves alternating between two optimisation steps:

i) Updating the rate matrix parameters $\theta$ by minimising the squared deviation of $\xi_t(x; R_t^\theta)$ from a baseline $c_t$, averaged over time and a chosen reference distribution $q_t(x)$

$$\theta \leftarrow \underset{\theta}{\text{argmin}}\int_0^1 \mathbb{E}_{q_t(x)}(\xi_t(x; R_t(\theta)) - c_t)^2\,dt$$

ii) Updating the baseline $c_t$ to match the expected value of $\xi_t$ under the true distribution $p_t(x)$

$$c_t \leftarrow \underset{c_t}{\text{argmin}}\,\mathbb{E}_{p_t(x)}(\xi_t(x; R_t^\theta) - c_t)^2$$

Alternatively, instead of treating $c_t$ as a scalar baseline, we can directly parametrize it as a neural network $c_t^\phi$. This allows us to jointly learn both $\theta$ and $\phi$ by solving the following objective:

$$\theta^*, \phi^* = \underset{\theta,\phi}{\text{argmin}}\,\mathbb{E}_{w(t),q_t(x)}\left[\partial_t\log\tilde{p}_t(x) - c_t^\phi - \sum_y R_t^\theta(x,y)\frac{p_t(y)}{p_t(x)}\right]^2, \tag{18}$$

which matches the Physics-Informed Neural Network (PINN) objective as derived in (Holderrieth et al., 2025, Proposition 6.1). At optimality, this objective recovers two important conditions: i) The learned network $c_t^{\phi'}$ recovers the true derivative $\partial_t\log Z_t$; and ii) The rate matrix $R_t^{\theta'}$ satisfies the Kolmogorov forward equation.

A key insight is that even though Equation (8) formally holds when the expectation is taken under $p_t$, the training objective in Equation (18) remains valid for any reference distribution $q_t$ as long as it shares support with $p_t$. This is because, at optimality, the residual

$$\partial_t\log\tilde{p}_t(x) - c_t^{\phi^*} - \sum_y R_t^{\theta^*}(x,y)\frac{p_t(y)}{p_t(x)} = 0, \forall x.$$

Integrating both sides with respect to $p_t$, and invoking the discrete Stein identity (Lemma 1), we find:

$$\mathbb{E}_{p_t}\left[\partial_t\log\tilde{p}_t(x) - c_t^{\phi^*}\right] = 0 \;\Rightarrow\; c_t^{\phi^*} = \mathbb{E}_{p_t}\left[\partial_t\log\tilde{p}_t(x)\right] = \partial_t\log Z_t.$$

This also naturally admits a coordinate ascent training procedure, where $\theta$ and $\phi$ are updated in turn:

$$\theta \leftarrow \underset{\theta}{\text{argmin}}\,\mathbb{E}_{w(t),q_t(x)}\left[\partial_t\log\tilde{p}_t(x) - c_t^\phi - \sum_y R_t^\theta(x,y)\frac{p_t(y)}{p_t(x)}\right]^2,$$

$$c_t^\phi \leftarrow \mathbb{E}_{w(t),q_t(x)}\left[\partial_t\log\tilde{p}_t(x) - \sum_y R_t^\theta(x,y)\frac{p_t(y)}{p_t(x)}\right],$$

In this light, the term the term $\sum_y R_t^\theta(x,y)\frac{p_t(y)}{p_t(x)}$ serves as a control variate for estimating $\partial_t \log Z_t$, effectively reducing variance in the learning signal. Under optimal training, $c_t^{\phi^*}$ accurately captures the log-partition derivative, confirming the correctness of the learned dynamics.

Since NDFS closely resembles LEAPS, we summarise the main distinctions below to highlight the unique contributions of our work:

- While LEAPS and DNFS yield similar objective functions, they are derived from different perspectives. DNFS derives the objective by learning the rate matrix to satisfy the Kolmogrove equation, whereas LEAPS learns the rate matrix by minimising the importance weights. Perhaps surprisingly, these two perspectives lead to similar objectives. However, the new perspective from the Kolmogorov equation offers a new insight for future research: leveraging more accurate estimators of $\partial_t \log Z_t$ to further improve performance, which is not evident from the LEAPS framework.

- The success of DNFS highly depends on the proposed Locally Equivariant Transformer (leTF). Compared to the locally equivariant networks in LEAPS, leTF offers greater model capacity and improved adaptability, making it more suitable for diverse modalities and complex input structures. We hope this architectural advancement will inspire future developments, which are essential for advancing both LEAPS and DNFS.

- Unlike LEAPS, which is only evaluated on synthetic Ising and Potts models, DNFS is tested on broader applications, including sampling from Ising models, training EBMs, and solving combinatorial optimisation problems. We hope this wider empirical scope will inspire further research into additional applications of discrete neural samplers.

## B  Derivation of Locally Equivariant Transformer

### B.1  Proof of Proposition 1

It is worth noting that Proposition 1 was first introduced in (Zhang et al., 2023b, Proposition 5), and subsequently utilized by Campbell et al. (2024) to construct the conditional rate matrix, as well as by Holderrieth et al. (2025) in the development of the locally equivariant network. For completeness, we provide a detailed proof of Proposition 1 in this section.

We begin by formally defining the one-way rate matrix

**Definition 1** (One-way Rate Matrix). *A rate matrix $R$ is one-way if and only if $R(y,x) > 0 \Rightarrow R(x,y) = 0$. In other words, if a one-way rate matrix permits a transition from $x$ to $y$, then the transition probability from $y$ to $x$ must be zero.*

We then restate Proposition 1 and provide a detailed proof as follows.

**Proposition 1.** *For a rate matrix $R_t$ that generates the probabilistic path $p_t$, there exists a one-way rate matrix $Q_t(y,x) = \left[R_t(y,x) - R_t(x,y)\frac{p_t(y)}{p_t(x)}\right]_+$ if $y \neq x$ and $Q_t(x,x) = \sum_{y \neq x} Q_t(y,x)$, that generates the same probabilistic path $p_t$, where $[z]_+ = \max(z,0)$ denotes the ReLU operation.*

*Proof.* We first prove that the one-way rate matrix $Q_t$ generates the same probabilistic path as $R_t$:

$$
\begin{aligned}
\sum_y Q_t(x,y)p_t(y) &= \sum_{y \neq x} Q_t(x,y)p_t(y) - Q_t(y,x)p_t(x) \\
&= \sum_{y \neq x} [R_t(x,y)p_t(y) - R_t(y,x)p_t(x)]_+ - [R_t(y,x)p_t(x) - R_t(x,y)p_t(y)]_+ \\
&= \sum_{y \neq x} R_t(x,y)p_t(y) - R_t(y,x)p_t(x) \\
&= \sum_y R_t(x,y)p_t(y) = \partial_t p_t(x),
\end{aligned}
$$

which completes the proof. We then show that $Q_t$ is one-way:

$$Q_t(y,x) > 0 \Rightarrow R_t(y,x) - R_t(x,y)\frac{p_t(y)}{p_t(x)} > 0 \Rightarrow R_t(x,y) - R_t(y,x)\frac{p_t(x)}{p_t(y)} < 0$$

$$\Rightarrow Q_t(x,y) = \left[ R_t(x,y) - R_t(y,x)\frac{p_t(x)}{p_t(y)} \right]_+ = 0. \tag{19}$$

Thus, $Q_t(y,x) > 0 \Rightarrow Q_t(x,y) = 0$, which completes the proof. $\qquad\square$

## B.2 Locally Equivariant Networks

Based on Proposition 1, we can parametrise $R_t^\theta$ as a one-way rate matrix, which is theoretically capable of achieving the optimum that minimizes Equation (5). Although the one-way rate matrix is a restricted subset of general rate matrices and thus offers limited flexibility, it enables efficient computation of the objective in Equation (5). To see this, we first formally define the local equivariant network, originally proposed in Holderrieth et al. (2025).

**Definition 2** (Locally Equivariant Network). *A neural network $G$ is locally equivariant if and only if*

$$G_t(\tau, i|x) = -G_t(x_i, i|Swap(x, i, \tau)), \quad i = 1, \ldots, d \tag{20}$$

*where $Swap(x, i, \tau) = (x_1, \ldots, x_{i-1}, \tau, x_{i+1}, \ldots, x_d)$ and $\tau \in \{1, \ldots, S\}$.*

We can then parametrise the one-way rate matrix $R_t$ using a locally equivariant network $G_t$: $R_t(\tau, i|x) = [G_t(\tau, i|x)]_+$, if $\tau \neq x_i$ and $R_t(x, x) = \sum_{y \neq x} R_t(y, x)$. This construction ensures that $R_t$ is a one-way rate matrix. To see this, consider a state $y = (x_1, \ldots, x_{i-1}, \tau, x_{i+1}, \ldots, x_d)$. If $R_t(y, x) > 0$, then

$$G_t(\tau, i|x) > 0 \Rightarrow -G_t(x_i, i|\text{Swap}(x, i, \tau)) > 0 \Rightarrow R_t(x, y) < 0, \tag{21}$$

demonstrating the one-way property. With this parameterisation, the objective function in Equation (5) can be computed as

$$\delta_t(x; R_t) = \partial_t \log p_t(x) + \sum_{i, y_i \neq x_i} R_t^\theta(y_i, i|x) - R_t^\theta(x_i, i|y)\frac{p_t(y)}{p_t(x)}$$

$$= \partial_t \log p_t(x) + \sum_{i, y_i \neq x_i} [G_t^\theta(y_i, i|x)]_+ - [G_t^\theta(x_i, i|y)]_+ \frac{p_t(y)}{p_t(x)}$$

$$= \partial_t \log p_t(x) + \sum_{i, y_i \neq x_i} [G_t^\theta(y_i, i|x)]_+ - [-G_t^\theta(y_i, i|x)]_+ \frac{p_t(y)}{p_t(x)},$$

where the final expression only requires a single forward pass of the network $G_t^\theta$ to compute the entire sum, significantly reducing computational cost. To construct a locally equivariant network, we first define its fundamental building block, the hollow network, as follows

**Definition 3** (Hollow Network). *Let $x_{i \leftarrow \tau} = (x_1, \ldots, x_i = \tau, \ldots, x_d) \in \mathcal{X}$ denote the input tokens with its $i$-th component set to $\tau$. A function $H : \mathcal{X} \mapsto \mathbb{R}^{d \times h}$ is called a hollow network if it satisfies the following condition*

$$H(x_{i \leftarrow \tau})_{i,:} = H(x_{i \leftarrow \tau'})_{i,:} \qquad \text{for all } \tau, \tau' \in \mathcal{S},$$

*where $M_{i,:}$ denotes the $i$-th row of the matrix $M$. In other words, the output at position $i$ is invariant to the input at position $i$; that is, the $i$-th input does not influence the $i$-th output.*

Inspired by Holderrieth et al. (2025), we then introduce the following proposition, which provides a concrete method for instantiating a locally equivariant network.

**Proposition 2** (Instantiation of Locally Equivariant Networks). *Let $x \in \mathcal{X}$ denote the input tokens and $H : \mathcal{X} \mapsto \mathbb{R}^{d \times h}$ be a hollow network. Furthermore, for each token $\tau \in \mathcal{S}$, let $\omega_\tau \in \mathbb{R}^h$ be a learnable projection vector. Then, the locally equivariant network can be constructed as:*

$$G(\tau, i|x) = (\omega_\tau - \omega_{x_i})^T H(x)_{i,:}.$$

*Proof.* We verify local equivariance by showing:

$$
\begin{aligned}
G(\tau, i|x) &= (\omega_\tau - \omega_{x_i})^T H(x)_{i,:} \\
&= -(\omega_{x_i} - \omega_\tau)^T H(\text{Swap}(x, i, \tau))_{i,:} \\
&= -G(x_i, i|\text{Swap}(x, i, \tau)),
\end{aligned}
$$

where the second equality follows from the definition of the hollow network in Definition 3, which ensures that the $i$-th output is invariant to the changes in the $i$-th input. This confirms that $G$ satisfies the required local equivariance condition as in Definition 2 $\qquad\square$

Based on Proposition 2, we present two locally equivariant architectures introduced in Holderrieth et al. (2025), followed by our proposed locally equivariant transformer.

**Locally Equivariant MLP (leMLP) (Holderrieth et al., 2025).** Let $x \in \mathbb{R}^{d \times h}$ be the embedded input data. To construct a locally equivariant multilinear perceptron (MLP), we first define a hollow MLP as $H_{\text{MLP}}(x) = \sum_{k=1}^{K} \sigma(W^k x + b^k)$ where each $W^k \in \mathbb{R}^{d \times d}$ is a weight matrix with zero diagonal entries (i.e., $W_{ii} = 0$ for all $i$), $b^k \in \mathbb{R}^h$ is the bias term, and $\sigma$ denotes an element-wise activation function. A locally equivariant MLP can then be defined as $G(\tau, i|x) = (\omega_\tau - \omega_{x_i})^T H_{\text{MLP}}(x)_{i,:}$.

**Locally Equivariant Attention (leAttn) (Holderrieth et al., 2025).** Let $x = (x_1, \ldots, x_d) \in \mathbb{R}^{d \times h}$ be the embedded input data. Similarly, we first define a hollow attention network as

$$
H_{\text{Attn}}(x)_{i,:} = \sum_{s \neq i} \frac{\exp(k(x_s)^T q(x_s))}{\sum_{t \neq i} \exp(k(x_t)^T q(x_t))} v(x_s),
$$

where $q, k, v$ denote the query, key, and value functions, respectively. Thus a locally equivariant attention network can be defined as $G(\tau, i|x) = (\omega_\tau - \omega_{x_i})^T H_{\text{Attn}}(x)_{i,:}$.

While these two architectures[3] offer concrete approaches for constructing locally equivariant networks, their flexibility is limited, as they each consider only a single layer. More importantly, naively stacking multiple MLP or attention layers violates local equivariance, undermining the desired property.

### B.3 Locally Equivariant TransFormer (leTF)

In this section, we present the implementation details of the proposed Locally Equivariant Transformer (leTF), an expressive network architecture designed to preserve local equivariance. As introduced in Section 3.3, leTF is formulated as

$$
G_t^\theta(\tau, i|x) = (\omega_\tau - \omega_{x_i})^T H_{\text{HTF}}(x)_{i,:}, \qquad (22)
$$

where $\omega$ denotes the learnable token embeddings produced by the projection layer (illustrated in Figure 2), and $H_{\text{HTF}}$ represents the Hollow Transformer module. In the following, we focus on the implementation details of the hollow transformer $H_{\text{HTF}}$.

As illustrated in Figure 8, the hollow transformer comprises $L$ bidirectional causal attention layers followed by a single attention readout layer. For clarity, we omit details of each causal attention layer, as they follow the standard Transformer architecture (Vaswani et al., 2017). We denote the outputs of the final left-to-right and right-to-left causal attention layers as

$$
Q_L, K_L, V_L = \text{CausalAttn}_{\text{L2R}}(\text{input});
$$
$$
Q_R, K_R, V_R = \text{CausalAttn}_{\text{R2L}}(\text{input}).
$$

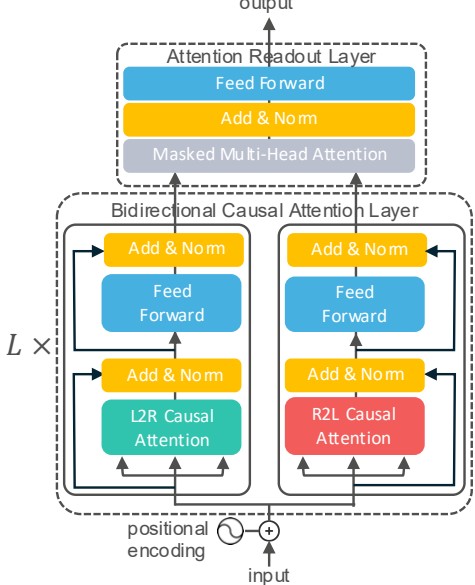

Figure 8: Illustration of the hollow transformer.

---

[3]Note that Holderrieth et al. (2025) also introduces a locally equivariant convolutional network; we refer interested readers to their work for further details.

---

**Algorithm 1** Training Procedure of DNFS

---

**Input**: initial rate matrix $R_t^\theta$, probability path $p_t$, time spans $\{t_k\}_{k=0}^K$, outer-loop batch size $M$, inner-loop batch size $N$, replay buffer $\mathcal{B}$

1: $\mathcal{B} \leftarrow \emptyset$          ▷ Initialise replay buffer
2: **while** Outer-Loop **do**
3:      $\{x_{t_k}^{(m)}\}_{m=1,k=0}^{M,K} \sim \overrightarrow{\mathbb{Q}}^{p_0, R_t^{\theta_{\mathrm{sg}}}}$          ▷ Generate training samples
4:      $c_t \leftarrow \frac{1}{M} \sum_m \partial_t \log \tilde{p}_t(x_t^{(m)}) - \sum_y R_t^{\theta_{\mathrm{sg}}}(x_t^{(m)}, y) \frac{p_t(y)}{p_t(x_t^{(m)})}, \forall t \in \{t_k\}_{k=0}^K$      ▷ $\partial_t \log Z_t$
5:      $\mathcal{B} \leftarrow \mathcal{B} \cup \{(t_k, x_{t_k}^{(m)})\}_{m=1,k=0}^{M,K}$          ▷ update replay buffer
6:      **while** Inner-Loop **do**
7:          $\{(t, x_t^{(n)})\}_{n=1}^N \sim \mathcal{U}(\mathcal{B})$          ▷ Uniformly sample from buffer
8:          $\xi_t^{(n)} \leftarrow \partial_t \log \tilde{p}_t(x_t^{(n)}) - \sum_y R_t^\theta(x_t^{(n)}, y) \frac{p_t(y)}{p_t(x_t^{(n)})}$
9:          $\mathcal{L}(\theta) \leftarrow \frac{1}{N} \sum_n (\xi_t^{(n)} - c_t)^2$          ▷ Compute training loss
10:          $\theta \leftarrow \text{optimizer\_step}(\theta, \nabla_\theta \mathcal{L}(\theta))$          ▷ Perform gradient update
11:      **end while**
12: **end while**

**Output**: trained rate matrix $R_t^\theta$

---

Because causal attention restricts each token to attend only to its preceding tokens, the resulting outputs $Q, K, V$ inherently satisfy the hollow constraint. In the readout layer, we first fuse the two query representations by computing $Q = Q_L + Q_R$, and then apply a masked multi-head attention to produce the final output

$$\text{softmax}\left(\frac{Q[K_L \odot M_L, K_R \odot M_R]^T}{\sqrt{2d_k}}\right)[V_L, V_R],$$

where $\odot$ denotes the element-wise product, $d_k$ is the dimensionality of the key vectors, and $M_L$ and $M_R$ masks that enforce the hollow constraint by masking out future-token dependencies in the left-to-right and right-to-left streams, respectively.

## C   Details of Training and Sampling of DNFS

### C.1   Training and Sampling Algorithms

The training and sampling procedures are presented in Algorithms 1 and 2. For clarity, the rate matrix $R_t^\theta$ is parametrised using the proposed locally equivariant network, defined as $R_t^\theta(y, x) \triangleq [G_t^\theta(y_i, i|x)]_+$, where $y$ and $x$ only differ at the $i$-th coordinate. To initiate training, we discretise the time interval $[0, 1]$ into a set of evenly spaced time spans $\{t_k\}_{k=0}^K$, satisfying $0 = t_0 < \cdots < t_K = 1$ and $2t_k = t_{k+1} + t_{k-1}$ for all valid indices $k$.

In each outer loop of training, we generate trajectory samples by simulating the forward process under the current model parameters. Specifically, samples are drawn from the probability path $\overrightarrow{\mathbb{Q}}^{p_0, R_t^{\theta_{\mathrm{sg}}}}$, defined by the initial distribution $p_0$ and the current rate matrix $R_t^{\theta_{\mathrm{sg}}}$, where $\theta_{\mathrm{sg}}$ denotes stop_gradient($\theta$). This forward trajectory is simulated using the Euler–Maruyama method, as detailed in Algorithm 2, and stored in a replay buffer for reuse. During the inner training loop, we draw mini-batches uniformly from the buffer and compute the training loss based on Equation (10). The model parameters are then updated via gradient descent, as described in steps 6–11 of Algorithm 1.

### C.2   Efficient Ratio Computation

Although the ratio $\left[\frac{p_t(y)}{p_t(x)}\right]_{y \in \mathcal{N}(x)}$ can be computed in parallel, it remains computationally expensive in general. However, in certain settings, such as sampling from Ising models and solving combinatorial optimisation problems, the ratio can be evaluated efficiently due to the specific form of the underlying distribution. In these cases, the unnormalized distribution takes a quadratic form:

$$p(x) \propto \tilde{p}(x) \triangleq \exp(x^T W x + h^T x), \quad x \in \{0, 1\}^d, W \in \mathbb{R}^{d \times d}, h \in \mathbb{R}^d, \tag{23}$$

---

**Algorithm 2** Sampling Procedure of DNFS

---

**Input**: trained rate matrix $R_t^\theta$, initial density $p_0$, # steps $K$

1: $x_0 \sim p_0$, $\quad \Delta t \leftarrow \frac{1}{K}$, $\quad t \leftarrow 0$ $\qquad\qquad\qquad\qquad\qquad\qquad$ ▷ Initialisation
2: **for** $k = 0, \dots, K - 1$ **do**
3: $\qquad x_{t+\Delta t} \leftarrow \text{Cat}(\mathbf{1}_{x_{t+\Delta t}=x} + R_t(x_{t+\Delta t}, x)\Delta t)$ $\qquad\qquad$ ▷ Euler-Maruyama update
4: $\qquad t \leftarrow t + \Delta t$
5: **end for**

**Output**: generated samples $x_1$

---

where the neighbourhood $\mathcal{N}(x)$ is defined as the 1-Hamming Ball around $x$, i.e., all vectors differing from $x$ in exactly one bit. Let $y$ be such a neighbor obtained by flipping the $i$-th bit: $y_i = \neg x_i$ and $y_j = x_j$ for $j \neq i$. The log of the unnormalised probability can be decomposed as:

$$\log \tilde{p}(x) = \sum_{a \neq i, b \neq i} x_a W_{ab} x_b + x_i \sum_{b \neq i} W_{ib} x_b + x_i \sum_{a \neq i} x_a W_{ai} + x_i W_{ii} x_i + \sum_a h_a x_a. \tag{24}$$

Thus, the change in log-probability when flipping bit $i$ is:

$$\log p(y) - \log p(x) = (1 - 2x_i) \left( \sum_{b \neq i} W_{ib} x_b + \sum_{a \neq i} x_a W_{ai} + W_{ii} + h_i \right). \tag{25}$$

This expression can be vectorised to efficiently compute the log-ratios for all neighbours:

$$\left[ \log \frac{p(y)}{p(x)} \right]_{y \in \mathcal{N}(x)} = (1 - 2x) \odot ((W + W^T)x - \text{diag}(W) + h), \tag{26}$$

where $\odot$ denotes element-wise multiplication. Furthermore, in special cases, such as combinatorial optimisation where $W$ is symmetric with zero diagonal, the log-ratio simplifies to

$$\left[ \log \frac{p(y)}{p(x)} \right]_{y \in \mathcal{N}(x)} = (1 - 2x) \odot (2Wx + h). \tag{27}$$

In more general settings, where $x$ is a categorical variable and the energy is non-quadratic, there is no closed-form solution for efficiently calculating the likelihood ratio. However, it can be approximated through a first-order Taylor expansion (Grathwohl et al., 2021). Specifically,

$$\log p_t(y) - \log p_t(x) = (y - x)\nabla_x \log p_t(x), \tag{28}$$

which gives the following approximation:

$$\left[ \log \frac{p_t(y)}{p_t(x)} \right]_{i,j} = [\nabla_x \log p_t(x)]_{i,j} - x_i^T [\nabla_x \log p_t(x)]_{i,:}, \tag{29}$$

where $\left[ \log \frac{p_t(y)}{p_t(x)} \right]_{i,j} \triangleq \log \frac{p_t(x_1, \dots, x_{i-1}, j, x_{i+1}, \dots, x_d)}{p_t(x)}$ and we take the fact that $y$ and $x$ differ only in one position. This approximation requires computing $\nabla_x \log p_t(x)$ just once to estimate the ratio for the entire neighbourhood, thus improving computational efficiency. However, it introduces bias into the training objective in Equation (5), potentially leading to suboptimal solutions.

To illustrate this, we train DNFS to sample from pre-trained deep EBMs by minimising the objective in Equation (5) using two methods for computing the ratio $\frac{p_t(y)}{p_t(x)}$: i) exact computation in parallel; and ii) an approximation via the Taylor expansion in Equation (29). As shown Figure 9, using the approximate ratio leads to inaccurate samples that tend to lie in overly smoothed regions of the energy landscape. This degradation in sample quality is likely caused by the bias introduced by the approximation. Consequently, in deep EBM settings where the energy function is non-quadratic, we opt to compute the exact ratio in parallel, leaving the development of more efficient and unbiased approximations for future work.

## D Applications to Discrete Neural Flow Samplers

In this section, we introduce details of two applications to discrete neural flow samplers: training discrete energy-based models and solving combinatorial optimisation problems.

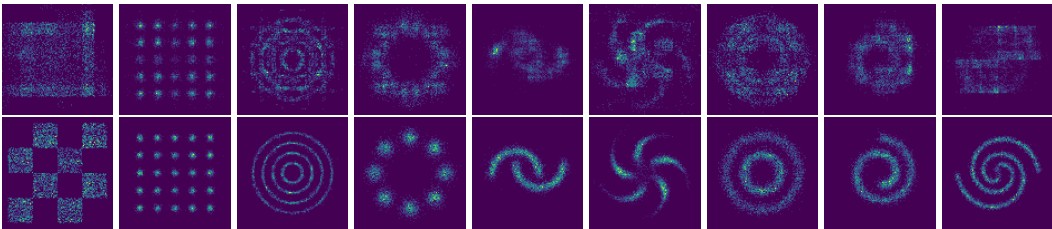

Figure 9: Illustration of the effect of ratio computation in Equation (5). Top row: results using the Taylor expansion approximation; bottom row: results using exact computation. The approximated ratio yields less accurate samples, likely due to the bias introduced in the training objective.

### D.1 Importance Sampling

We begin by reviewing the fundamentals of importance sampling, followed by a proof based on the Radon-Nikodym derivative. This approach was also introduced in Lee et al. (2025); Holderrieth et al. (2025); Pani et al. (2025). For completeness, we include a brief recap to make the paper more self-contained.

**Importance Sampling.** Consider a target distribution $\pi(x) = \frac{\rho(x)}{Z}$, where $\rho(x) \geq 0$ is the unnormalised distribution and $Z = \sum_x \rho(x)$ denotes the normalising constant, which is typically intractable. For a test function $\phi(x)$ of interest, estimating its expectation under $\pi$ through direct sampling can be challenging. Importance Sampling (IS) (Kahn, 1950) addresses this by introducing a proposal distribution $q(x)$ that is easier to sample from. The expectation under $\pi$ can then be re-expressed as

$$\mathbb{E}_{\pi(x)}[\phi(x)] = \frac{1}{Z}\mathbb{E}_{q(x)}\left[\frac{\rho(x)}{q(x)}\phi(x)\right] = \frac{\mathbb{E}_{q(x)}\left[\frac{\rho(x)}{q(x)}\phi(x)\right]}{\mathbb{E}_{q(x)}\left[\frac{\rho(x)}{q(x)}\right]}. \tag{30}$$

This leads to the Monte Carlo estimator:

$$\mathbb{E}_{\pi(x)}[\phi(x)] \approx \sum_{k=1}^{K} \frac{w^{(k)}}{\sum_{j=1}^{K} w^{(j)}}\phi(x^{(k)}), \quad x^{(k)} \sim q(x), \tag{31}$$

where $w^{(k)} = \frac{\rho(x^{(k)})}{q(x^{(k)})}$ denotes the importance weight. Although this estimator is consistent as $K \to \infty$, it often suffers from high variance and low effective sample size (Thiébaux & Zwiers, 1984), especially when the proposal $q$ is poorly matched to the target $\pi$. In theory, the variance of the estimator is minimized when $q(x) \propto \rho(x)\phi(x)$, yielding a zero-variance estimator. While this condition is rarely attainable in practice, it provides a useful guideline: a well-designed proposal should closely approximate the target distribution, i.e., $q(x) \approx \pi(x)$.

**CTMT-Inspired Importance Sampling.** As noted previously, an ideal proposal should should closely approximate the target, i.e., $q(x) \approx \pi(x)$. This motivates the use of continuous-time Markov chains (CTMCs) to construct the proposal. Specifically, let $R_t(y, x)$ denote a rate matrix that defines a forward CTMC with initial distribution $p_0 \propto \eta$, generating a probability path denoted by $\overrightarrow{\mathbb{Q}}^{\eta,R_t}$. To complement this, we define a backward CTMC with initial distribution $p_1 \propto \rho$ and interpolated marginals $p_t \propto \tilde{p}_t := \rho^t \eta^{1-t}$. The backward process is governed by the rate matrix $R'_t(y, x) = R_t(x, y)\frac{p_t(y)}{p_t(x)}$, leading to the backward path distribution $\overleftarrow{\mathbb{Q}}^{\rho,R'_t}$. This construction yields the following importance sampling identity:

$$\mathbb{E}_{\pi(x)}[\phi(x)] = \mathbb{E}_{x_{0\leq t\leq 1}\sim\overleftarrow{\mathbb{Q}}^{\rho,R'_t}}[\phi(x_1)] = \mathbb{E}_{x_{0\leq t\leq 1}\sim\overrightarrow{\mathbb{Q}}^{\eta,R_t}}\left[\frac{\overleftarrow{\mathbb{Q}}^{\rho,R'_t}(x_{0\leq t\leq 1})}{\overrightarrow{\mathbb{Q}}^{\eta,R_t}(x_{0\leq t\leq 1})}\phi(x_1)\right], \tag{32}$$

which can be approximated via Monte Carlo sampling:

$$\mathbb{E}_{\pi(x)}[\phi(x)] \approx \sum_{k=1}^{K} \frac{w^{(k)}}{\sum_{j=1}^{K} w^{(j)}}\phi(x_1^{(k)}), \quad x_{0\leq t\leq 1}^{(k)} \sim \overrightarrow{\mathbb{Q}}^{\eta,R_t}, \quad w^{(k)} = \frac{\overleftarrow{\mathbb{Q}}^{\rho,R'_t}(x_{0\leq t\leq 1}^{(k)})}{\overrightarrow{\mathbb{Q}}^{\eta,R_t}(x_{0\leq t\leq 1}^{(k)})}. \tag{33}$$

This estimator is consistent for any choice of $R_t$, and it becomes zero-variance when $R_t$ satisfies the Kolmogorov equation in Equation (4), a condition that can be approximately enforced by minimizing the loss in Equation (5). Before delving into the computation of the importance weights $\frac{\overleftarrow{\mathbb{Q}}^{\rho,R'_t}}{\overrightarrow{\mathbb{Q}}^{\eta,R_t}}$, we introduce two key lemmas that underpin the derivation.

**Lemma 3** (Radon-Nikodym Derivative (Del Moral & Penev, 2017)). *Let $p_0$ and $p_t$ be two initial distributions; $R_s$ and $R'_s$ be two rate matrices, which induce the forward and backward CTMCs $\overrightarrow{\mathbb{Q}}^{p_0,R}$ and $\overleftarrow{\mathbb{Q}}^{p_t,R'}$ over the time interval $[0,t]$, respectively. Then,*

$$\log \frac{\overleftarrow{\mathbb{Q}}^{R'_t|p_t}}{\overrightarrow{\mathbb{Q}}^{R_t|p_0}} = \int_0^t R'_s(x_s,x_s) - R_s(x_s,x_s)\,\mathrm{d}s + \sum_{s,x_s^-\neq x_s} \log \frac{R'_s(x_s^-,x_s)}{R_s(x_s,x_s^-)}, \tag{34}$$

*which induces that*

$$\frac{\overleftarrow{\mathbb{Q}}^{p_t,R'_t}}{\overrightarrow{\mathbb{Q}}^{p_0,R_t}} = \frac{p_t(x_t)}{p_0(x_0)} \frac{\exp\left(\int_0^t R'_s(x_s,x_s)\,\mathrm{d}s\right)}{\exp\left(\int_0^t R_s(x_s,x_s)\,\mathrm{d}s\right)} \prod_{s,x_s^-\neq x_s} \frac{R'_s(x_s^-,x_s)}{R_s(x_s,x_s^-)}. \tag{35}$$

*Proof.* For a comprehensive proof, we refer readers to (Campbell et al., 2024, Appendix C.1), which follows the exposition in Del Moral & Penev (2017). $\square$

**Lemma 4** (Fundamental Theorem of Calculus). *Let $f : [0,T] \to \mathbb{R}$ be a piecewise differentiable function on the interval $[0,T]$. Suppose that $f$ is differentiable except at a finite set of discontinuity points $\{s_i\}_{i=1}^n \subset [0,T]$, where the left-hand limit $f(s_i^-)$ and the right-hand limit $f(s_i^+)$ at each $s_i$ exist but are not necessarily equal. Then, the total change of $f$ can be expressed as*

$$f(t) - f(0) = \int_0^t f'(s)\,\mathrm{d}s + \sum_{s\in\{s_i\}_{i=1}^n \cap (0,t)} \left[f(s^+) - f(s^-)\right], \tag{36}$$

*for all $t \in [0,T]$, where $f'(s)$ denotes the derivative of $f$ at points where $f$ is differentiable.*

Now, it is ready to compute the importance weight. Specifically,

$$\begin{aligned}
\log \frac{\overleftarrow{\mathbb{Q}}^{p_t,R'_t}}{\overrightarrow{\mathbb{Q}}^{p_0,R_t}} &= \log \frac{\tilde{p}_t}{\tilde{p}_0} + \log \frac{Z_0}{Z_t} + \int_0^t R'_s(x_s,x_s) - R_s(x_s,x_s)\,\mathrm{d}s + \sum_{s,x_s^-\neq x_s} \log \frac{R'_s(x_s^-,x_s)}{R_s(x_s,x_s^-)} \\
&= \log \frac{Z_0}{Z_t} + \int_0^t \partial_s \log \tilde{p}_s\,\mathrm{d}s + \int_0^t R'_s(x_s,x_s) - R_s(x_s,x_s)\,\mathrm{d}s + \sum_{s,x_s^-\neq x_s} \log \frac{R'(x_s^-,x_s)p_s(x_s)}{R(x_s,x_s^-)p_s(x_s^-)} \\
&= \log \frac{Z_0}{Z_t} + \int_0^t \partial_s \log \tilde{p}_s\,\mathrm{d}s + \int_0^t R'_s(x_s,x_s) - R_s(x_s,x_s)\,\mathrm{d}s \\
&= \log \frac{Z_0}{Z_t} + \int_0^t \partial_s \log \tilde{p}_s\,\mathrm{d}s + \int_0^t -\sum_{y\neq x_s} R_s(x_s,y)\frac{p_s(y)}{p_s(x_s)} - R_s(x_s,x_s)\,\mathrm{d}s \\
&= \log \frac{Z_0}{Z_t} + \int_0^t \partial_s \log \tilde{p}_s - \sum_y R_s(x_s,y)\frac{p_s(y)}{p_s(x_s)}\,\mathrm{d}s,
\end{aligned}$$

where the first equation follows from Lemma 3, and the second from Lemma 4, leveraging the fact that $t \mapsto \log \tilde{p}_t$ is piecewise differentiable. The final equality holds by the detailed balance condition satisfied by the backward rate matrix, namely $R'_t(y,x)p_t(x) = R_t(x,y)p_t(y)$. Importantly, although the partition function $Z_t$ is generally intractable, it cancels out in practice through the use of the self-normalised importance sampling estimator as in Equation (32).

**Free Energy and Internal Energy Estimation.** To estimate the log-partition function $\log Z_t$, we consider the following lower bound

$$\log Z_t = \log \mathbb{E}_{p_t}\left[\frac{Z_t}{Z_0}\right] = \log \mathbb{E}_{x_{0\leq s\leq t}\sim\overrightarrow{\mathbb{Q}}^{p_0,R_t}}\left[\frac{\overleftarrow{\mathbb{Q}}^{p_t,R'_t}}{\overrightarrow{\mathbb{Q}}^{p_0,R_t}}\frac{Z_t}{Z_0}\right]$$

$$\geq \mathbb{E}_{x_{0\leq s\leq t}\sim\overrightarrow{\mathbb{Q}}^{p_0,R_t}}\left[\log\frac{\overleftarrow{\mathbb{Q}}^{p_t,R'_t}}{\overrightarrow{\mathbb{Q}}^{p_0,R_t}} + \log\frac{Z_t}{Z_0}\right]$$

$$= \mathbb{E}_{x_{0\leq s\leq t}\sim\overrightarrow{\mathbb{Q}}^{p_0,R_t}}\left[\int_0^t \partial_s \log\tilde{p}_s(x_s) - \sum_y R_s(x_s,y)\frac{p_s(y)}{p_s(x_s)}\,\mathrm{d}s\right], \tag{37}$$

where we assume that $Z_0 = 1$. Considering the test function $\phi = \log\tilde{p}_t$, one can also estimate the negative entropy, which is related to the internal energy:

$$\mathbb{E}_{p_t}[\log\tilde{p}_t] = \mathbb{E}_{x_{0\leq s\leq t}\sim\overrightarrow{\mathbb{Q}}^{p_0,R_t}}\left[\frac{\overleftarrow{\mathbb{Q}}^{p_t,R'_t}}{\overrightarrow{\mathbb{Q}}^{p_0,R_t}}\log\tilde{p}_t(x_t)\right] \approx \sum_{k=1}^K \frac{\exp(w_t^{(k)})}{\sum_{j=1}^K \exp(w_t^{(j)})}\log\tilde{p}_t(x_t^{(k)}), \tag{38}$$

where $x_{0\leq s\leq t}^{(k)} \sim \overrightarrow{\mathbb{Q}}^{\eta,R_t}$, and $w_t^{(k)}(x_{0\leq s\leq t}^{(k)}) = \int_0^t \partial_s \log\tilde{p}_s(x_s^{(k)}) - \sum_y R_s(x_s^{(k)},y)\frac{p_s(y)}{p_s(x_s^{(k)})}\,\mathrm{d}s$.

**Effective Sample Size (ESS).** The Effective Sample Size (ESS) (Liu & Liu, 2001, Chapter 2) quantifies how many independent and equally weighted samples a set of importance-weighted Monte Carlo samples is effectively worth. It reflects both the quality and diversity of the weights: when most weights are small and a few dominate, the ESS is low, indicating that only a small subset of samples contributes meaningfully to the estimate. Formally, consider the importance sampling estimator:

$$\mathbb{E}_p[\phi(x)] \approx \sum_{k=1}^K \frac{\exp(w^{(k)})}{\sum_{j=1}^K \exp(w^{(j)})}\phi(x^{(k)}), \quad w^{(k)} = \log\frac{p(x^{(k)})}{q(x^{(k)})}, \quad x^{(k)} \sim q(x), \tag{39}$$

where $q$ is the proposal distribution. The normalised ESS is given by:

$$\mathrm{ESS} = \frac{1}{K}\frac{(\sum_{k=1}^K \exp(w^{(k)}))^2}{\sum_{k=1}^K \exp(2w^{(k)})} = \frac{1}{K\sum_{k=1}^K (\tilde{w}^{(k)})^2}, \tag{40}$$

where $\tilde{w}^{(k)} = \frac{\exp(w^{(k)})}{\sum_{j=1}^K \exp(w^{(j)})}$ denotes the normalised importance weight. In CTMC-based importance sampling, the estimator takes the form:

$$\mathbb{E}_{p_t}[\phi(x_t)] \approx \sum_{k=1}^K \frac{\exp(w_t^{(k)})}{\sum_{j=1}^K \exp(w_t^{(j)})}\phi(x_t^{(k)}), \quad x_{0\leq s\leq t}^{(k)} \sim \overrightarrow{\mathbb{Q}}^{\eta,R_t}, \tag{41}$$

with log-weight $w_t^{(k)} = \int_0^t \partial_s \log\tilde{p}_s(x_s^{(k)}) - \sum_y R_s(x_s^{(k)},y)\frac{p_s(y)}{p_s(x_s^{(k)})}\,\mathrm{d}s$. Thus, the corresponding ESS is computed as

$$\mathrm{ESS} = \frac{1}{K\sum_{k=1}^K (\tilde{w}_t^{(k)})^2}, \quad \tilde{w}_t^{(k)} = \frac{\exp(w_t^{(k)})}{\sum_{j=1}^K \exp(w_t^{(j)})}. \tag{42}$$

**Log-Likelihood Estimation.** Let $R_t^\theta(y,x) \triangleq R_t^\theta(\tau,i|x) = [G_t^\theta(\tau,i|x)]_+$ be a learned rate matrix, where $y = (x_1,\ldots,x_{i-1},\tau,x_{i+1},\ldots,x_d)$. This matrix parameterizes a forward that progressively transforms noise into data via Euler-Maruyama discretization:

$$x_{t+\Delta t} \sim \mathrm{Cat}\left(\mathbf{1}_{x_{t+\Delta t}=x_t} + R_t^\theta(x_{t+\Delta t},x_t)\Delta t\right), \quad x_0 \sim p_0. \tag{43}$$

The corresponding reverse-time rate matrix is given by

$$R_t^{\theta'}(y,x) = R_t^\theta(x,y)\frac{p_t(y)}{p_t(x)} = [G_t^\theta(x_i,i|y)]_+\frac{\tilde{p}_t(y)}{\tilde{p}_t(x)} = [G_t^\theta(y_i,i|x)]_+\frac{\tilde{p}_t(y)}{\tilde{p}_t(x)}, \tag{44}$$

which enables simulating the reverse CTMC that maps data back into noise:

$$x_{t-\Delta t} \sim \mathrm{Cat}\left(\mathbf{1}_{x_{t-\Delta t}=x_t} + R_t^\theta(x_t, x_{t-\Delta t})\frac{\tilde{p}_t(x_{t-\Delta t})}{\tilde{p}_t(x_t)}\Delta t\right), \quad x_1 \sim p_1. \tag{45}$$

This reverse process enables estimation of a variational lower bound (ELBO) on the data log-likelihood:

$$\log \mathbb{E}_{p_{\mathrm{data}}}[p_\theta(x)] = \log \mathbb{E}_{x_1 \sim p_1, x_{0 \leq t < 1} \sim \overleftarrow{\mathbb{Q}}^{R_t^{\theta'}|x_1}}\left[\frac{\overrightarrow{\mathbb{Q}}^{R_t^\theta|x_0}}{\overleftarrow{\mathbb{Q}}^{R_t^{\theta'}|x_1}}p_0(x_0)\right]$$

$$\geq \mathbb{E}_{x_1 \sim p_1, x_{0 \leq t < 1} \sim \overleftarrow{\mathbb{Q}}^{R_t^{\theta'}|x_1}}\left[\log \frac{\overrightarrow{\mathbb{Q}}^{R_t^\theta|x_0}}{\overleftarrow{\mathbb{Q}}^{R_t^{\theta'}|x_1}} + \log p_0(x_0)\right]$$

$$\approx \frac{1}{K}\sum_{k=1}^K \log \frac{\overrightarrow{\mathbb{Q}}^{R_t^\theta|x_0^{(k)}}}{\overleftarrow{\mathbb{Q}}^{R_t^{\theta'}|x_1^{(k)}}} + \log p_0(x_0^{(k)}), \quad x_{0 \leq t \leq 1}^{(k)} \sim p_1(x_1)\overleftarrow{\mathbb{Q}}^{R_t^{\theta'}|x_1},$$

where the log-ratio $\log \frac{\overrightarrow{\mathbb{Q}}^{R_t^\theta|x_0}}{\overleftarrow{\mathbb{Q}}^{R_t^{\theta'}|x_1}}$ can be evaluated using Lemma 3

$$\log \frac{\overrightarrow{\mathbb{Q}}^{R_t^\theta|x_0}}{\overleftarrow{\mathbb{Q}}^{R_t^{\theta'}|x_1}} = \int_1^0 R_s^{\theta'}(x_s, x_s) - R_s^\theta(x_s, x_s)\,\mathrm{d}s + \sum_{s, x_s^- \neq x_s} \log \frac{R_s^\theta(x_s, x_s^-)}{R_t^{\theta'}(x_s^-, x_s)}$$

$$= \int_1^0 \sum_{y \neq x_s}\left[R_s^\theta(y, x_s) - R_s^\theta(x_s, y)\frac{p_s(y)}{p_s(x_s)}\right]\mathrm{d}s + \sum_{s, x_s^- \neq x_s} \log \frac{p_s(x_s)}{p_s(x_s^-)}$$

$$= \int_1^0 \sum_{\substack{i=1,\ldots,d \\ y \in N(x), y_i \neq x_i}} [G_s^\theta(y_i, i|x_s)]_+ - [-G_s^\theta(y_i, i|x_s)]\frac{\tilde{p}_s(y)}{\tilde{p}_s(x_s)}\,\mathrm{d}s + \sum_{s, x_s^- \neq x_s} \log \frac{\tilde{p}_s(x_s)}{\tilde{p}_s(x_s^-)}.$$

## D.2 Training Discrete EBMs with Importance Sampling

To train a discrete EBM $p_\phi(x) \propto \exp(-E_\phi(x))$, we employ contrastive divergence, which estimates the gradient of the log-likelihood as

$$\nabla_\phi \mathbb{E}_{p_{\mathrm{data}}(x)}[\log p_\phi(x)] = \mathbb{E}_{p_\phi(x)}[\nabla_\phi E_\phi(x)] - \mathbb{E}_{p_{\mathrm{data}}(x)}[\nabla_\phi E_\phi(x)], \tag{46}$$

where the second term can be easily approximated using the training data with Monte Carlo estimation. To estimate the intractable expectation over $p_\phi$, MCMC method is typically used. However, for computational efficiency, only a limited number of MCMC steps are performed, resulting in a biased maximum likelihood estimator and suboptimal energy function estimates (Nijkamp et al., 2020). To address this issue, we replace MCMC with the proposed discrete neural flow samplers. Specifically, we train a rate matrix $R_t^\theta$ to sample from the target EBM $p_\phi$. The expectation over $p_\phi$ can then be estimated using CTMT-inspired importance sampling, as described in Appendix D.1:

$$\mathbb{E}_{p_\phi(x)}[\nabla_\phi E_\phi(x)] = \mathbb{E}_{x \sim \overrightarrow{\mathbb{Q}}^{p_0, R_t^\theta}}\left[\frac{\overleftarrow{\mathbb{Q}}^{p_\phi, R_t^{\theta'}}}{\overrightarrow{\mathbb{Q}}^{p_0, R_t^\theta}}\nabla_\phi E_\phi(x)\right] \approx \sum_{k=1}^K \frac{\exp(w^{(k)})}{\sum_{j=1}^K \exp(w^{(j)})}\nabla_\phi E_\phi(x^{(k)}),$$

where $w^{(k)} = \int_0^1 \xi_t(x_t; R_t^\theta)\,\mathrm{d}t$. To summarise, we jointly train the EBM $p_\phi$ and the DNFS by alternating the following two steps until convergence:

1) Updating the rate matrix parameters $\theta$ using the training procedure described in Algorithm 1;
2) Updating EBM $p_\phi$ via contrastive divergence, as defined in Equation (46).

## D.3 Combinatorial Optimisation as Sampling

Consider a general combinatorial optimisation problem of the form $\min_{x \in \mathcal{X}} f(x)$ subject to $c(x) = 0$. This problem can be reformulated as sampling from an unnormalised distribution $p(x) \propto \exp(-E(x)/T)$, where the energy function is defined as $E(x) = f(x) + \lambda c(x)$ (Sun et al.,

2023b). As the temperature $T \to \infty$, $p(x)$ approaches the uniform distribution over $\mathcal{X}$, while as $T \to 0$, $p(x)$ concentrates on the optimal solutions, becoming uniform over the set of minimisers. In this paper, we focus on two combinatorial optimisation problems: Maximum Independent Set (MIS) and Maximum Cut (MaxCut). Below, we define the energy functions used for each task following Sun et al. (2022b). Given a graph $G = (V, E)$, we denote its adjacency matrix by $A$, which is a symmetric and zero-diagonal binary matrix.

**MIS.** The MIS problem can be formulated as the following constrained optimisation task:

$$\min_{x \in \{0,1\}^d} -\sum_{i=1}^{|V|} x_i, \quad \text{s.t. } x_i x_j = 0, \forall (i, j) \in E. \tag{47}$$

We define the corresponding energy function in quadratic form as:

$$-\log p(x) \propto E(x) = -1^T x + \lambda \frac{x^T A x}{2}. \tag{48}$$

Thus, the log-probability ratio between neighboring configurations $y \in \mathcal{N}(x)$, differing from $x$ by a single bit flip, has a closed-form expression

$$\left[\log \frac{p(y)}{p(x)}\right]_{y \in \mathcal{N}(x)} = (1 - 2x) \odot (x - \frac{1}{2}\lambda A x). \tag{49}$$

Following Sun et al. (2022b), we set $\lambda = 1.0001$. After inference, we apply a post-processing step to ensure feasibility: we iterate over each node $x_i$, and if any neighbour $x_j = 1$ for $(x_i, x_j) \in E$, we set $x_j \leftarrow 0$. This guarantees that the resulting configuration $x$ is a valid independent set.

**Maxcut.** The Maxcut problem can be formulated as

$$\min_{x \in \{-1,1\}^d} -\sum_{i,j \in E} A_{i,j} \left(\frac{1 - x_i x_j}{2}\right). \tag{50}$$

We define the corresponding energy function as

$$-\log p(x) \propto E(x) = \frac{1}{4} x^T A x. \tag{51}$$

This leads to the following closed-form expression for the log-ratio

$$\left[\log \frac{p(y)}{p(x)}\right]_{y \in \mathcal{N}(x)} = -\frac{1}{2} A x \tag{52}$$

Since any binary assignment yields a valid cut, no post-processing is required for MaxCut.

### D.4 Locally Equivariant GraphFormer (leGF)

To train an amortized version of DNFS for solving combinatorial optimization problems, it is essential to condition the model on the underlying graph structure. To this end, we integrate Graphormer (Ying et al., 2021) into our proposed Locally Equivariant Transformer (leTF), resulting in the Locally Equivariant Graphformer (leGF).

The leGF architecture largely follows the structure of leTF, with the key difference being the computation of attention weights, which are modified to incorporate graph-specific structural biases. Following Ying et al. (2021), given a graph $G = (V, E)$, we define $\psi(v_i, v_j)$ as the shortest-path distance between nodes $v_i$ and $v_j$ if a path exists; otherwise, we assign it a special value (e.g., $-1$). Each possible output of $\psi$ is associated with a learnable scalar $b_{\psi(v_i, v_j)}$, which serves as a structural bias term in the self-attention mechanism. Let $A_{i,j}$ denote the $(i, j)$-th

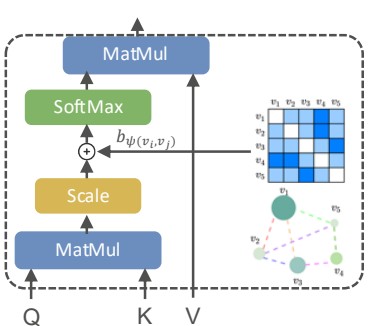

Figure 10: Illustration of the graph-aware attention mechanism. The figure is adapted from (Ying et al., 2021, Figure 1)

.

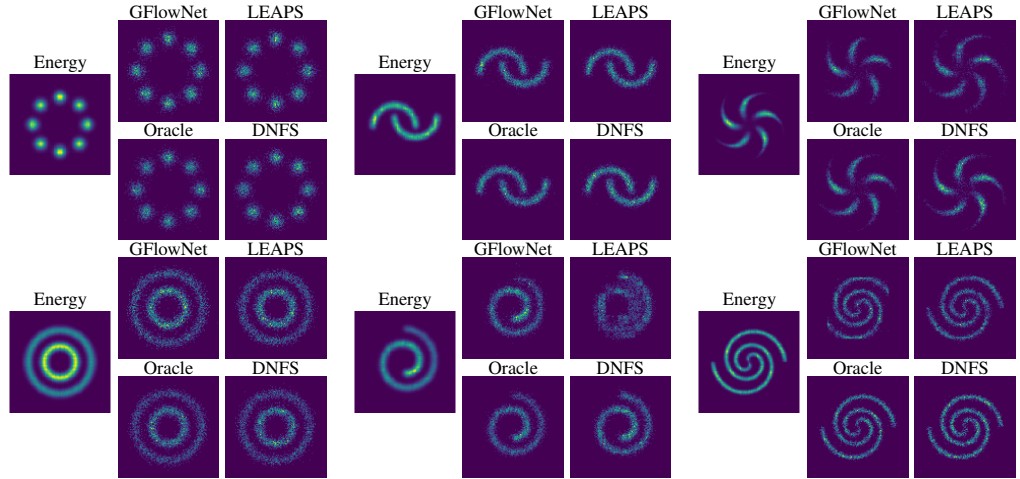

Figure 11: Additional visualisation of sampling from pre-trained EBMs.

element of the Query-Key interaction matrix. The attention weights are then computed as:

$$A_{i,j} = \text{softmax}(\frac{Q_i^T K_j}{\sqrt{d_k}} + b_{\psi(v_i, v_j)}),$$

where $b_{\psi(v_i, v_j)}$ is shared across all attention layers. An illustration of this graph-aware attention mechanism is shown in Figure 10. For further details, we refer the reader to Ying et al. (2021).

# E  Details of Experimental Settings and Additional Results

In this section, we present the detailed experimental settings and additional results. All experiments are conducted on a single Nvidia RTX A6000 GPU.

## E.1  Sampling from Unnormalised Distributions

### E.1.1  Experimental Details

**Sampling from Pre-trained EBMs.** In this experiment, we adopt energy discrepancy[4] to train an EBM, implemented as a 4 layer MLP with 256 hidden units and Swish activation. Once trained, the pretrained EBM serves as the target unnormalized distribution, with the initial distribution $p_0$ set to uniform. A probability path is then constructed using a linear schedule with 128 time steps. To parameterise the rate matrix, we employ the proposed locally equivariant transformer, in which the causal attention block consists of 3 multi-head attention layers, each with 4 heads and 128 hidden units. The model is trained using the AdamW optimizer with a learning rate of 0.0001 and a batch size of 128 for 1,000 epochs (100 steps per epoch). To prevent numerical instability from exploding loss, the log-ratio term $\log \frac{p_t(y)}{p_t(x)}$ is clipped to a maximum value of 5.

**Sampling from Ising Models.** We follow the experimental setup described in (Grathwohl et al., 2021, Section F.1). The energy function of the Ising model is given by $\log p(x) \propto a x^T J x + b^T x$. In Figure 5, we set $a = 0.1$ and $b = 0$. The probability path is constructed using a linear schedule with 64 time steps, starting from a uniform initial distribution. The leTF model comprises 3 bidirectional causal attention layers, each with 4 heads and 128 hidden units. Training is performed using the AdamW optimizer with a learning rate of 0.001, a batch size of 128, and for 1,000 epochs (100 steps per epoch). To prevent numerical instability, the log-ratio term is clipped to a maximum value of 5.

**Experimental Setup for Figures 1 and 3.** In this experiment, we set $a = 0.1$ and $b = 0.2$. The probability path is defined via a linear schedule over 64 time steps, starting from a uniform initial distribution. The leTF model is composed of 3 bidirectional causal attention layers, each with 4 heads

---

[4]https://github.com/J-zin/discrete-energy-discrepancy/tree/density_estimation

and 64 hidden units. Training is conducted using the AdamW optimiser with a learning rate of 0.001, a batch size of 128, and for 500 epochs (100 steps per epoch). To mitigate numerical instability, the log-ratio term is clipped to a maximum value of 5.

### E.1.2 Additional Results

**Additional Results of Sampling from Pre-trained EBMs.** We present additional results comparing DNFS to baseline methods on sampling from pre-trained EBMs in Figure 11. The results demonstrate that DNFS produces samples that closely resemble those from the oracle distribution.

**LEAPS with leConv and leTF.** While leConv performs comparably to DNFS on Ising models, its convolutional architecture may struggle to adapt to non-grid data structures. In this experiment, we use the LEAPS[5] algorithm (Holderrieth et al., 2025) to train a neural sampler with two different locally equivariant architectures: leConv (Holderrieth et al., 2025) and leTF (ours). Given the data has 32 dimensions, we pad it to 36 and reshape it into a $6 \times 6$ grid to make it compatible with leConv. As shown in Figure 12, LEAPS with leConv fails to achieve meaningful performance, whereas the proposed transformer-based architecture leTF performs comparably to DNFS. This result highlights the limitations of leConv, whose expressiveness is constrained on non-grid data, while leTF offers greater flexibility and generalisation across diverse input structures.


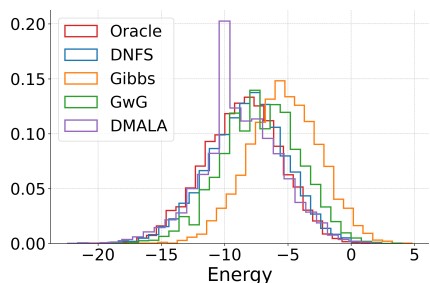

Figure 12: Comparison of leConv and leTF on sampling from pre-trained EBMs with LEAPS.

**Comparing to MCMC Methods.** Compared to MCMC methods, a key advantage of neural samplers is their ability to guarantee convergence when trained optimally, whereas MCMC methods often suffer from slow mixing and poor convergence. To highlight this benefit, we adopt the same setting as in Figure 5, using DNFS with 64 sampling steps. For a fair comparison, we evaluate MCMC baselines with the same number of steps, including Gibbs sampling (Casella & George, 1992), Gradient with Gibbs (GwG) (Grathwohl et al., 2021), and the Discrete Metropolis-adjusted Langevin Algorithm (DMALA) (Zhang et al., 2022b). We present energy histograms based on 5,000 samples in Figure 13, with a long-run Gibbs sampler serving as the oracle. The results show that short-run MCMC

Figure 13: Histogram of sample energy for different sampling methods.

methods struggle to produce accurate samples, although gradient-based variants (GwG and DMALA) outperform conventional Gibbs sampling. In contrast, the energy distribution produced by DNFS closely matches the oracle, demonstrating the effectiveness of the proposed neural sampler.

**Training Loss Comparison.** We train our model by minimising the loss function defined in Equation (5), where a smaller loss value reflects a better fit to the parameterised rate matrix. As such, the loss value serves as a proxy for training quality and model convergence. In Figure 14, we compare the training loss across different locally equivariant network architectures. The results show that the proposed locally equivariant transformer achieves the lowest loss value, indicating its superior capacity to fit the target

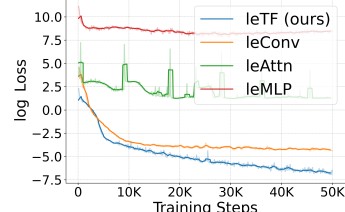
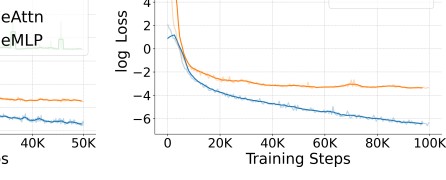

Figure 14: Training loss of different locally equivariant networks (ref: Figure 3).

Figure 15: Training loss comparison between DNFS and LEAPS (ref: Figure 5).

[5]https://github.com/malbergo/leaps

Table 2: Comparison of the estimated free energy $\mathcal{F}/D$, internel energy $\mathcal{E}/D$, entropy $\mathcal{S}/D$, and effective sample size for Ising models on $D = 10 \times 10$ grids at different temperatures.

| | Method | ESS | Free Energy $\mathcal{F}/D$ | Internal Energy $\mathcal{E}/D$ | Entropy $\mathcal{S}/D$ |
|---|---|---|---|---|---|
| $\sigma = 0.1$ | Optimal Value | 1 | $-3.6727$ | $-0.4282$ | $0.6489$ |
| | LEAPS | $0.9956 \pm 0.0001$ | $-3.6709 \pm 4.9182e^{-5}$ | $-0.4262 \pm 0.0034$ | $0.6489 \pm 0.0006$ |
| | DNFS | $0.9985 \pm 4.6912e^{-5}$ | $-3.6709 \pm 6.2463e^{-5}$ | $-0.4271 \pm 0.0010$ | $0.6488 \pm 0.0002$ |
| $\sigma = 0.22305$ | Optimal Value | 1 | $-2.1242$ | $-1.4763$ | $0.2855$ |
| | LEAPS | $0.3631 \pm 0.1184$ | $-2.1011 \pm 0.0002$ | $-1.4493 \pm 0.0130$ | $0.2873 \pm 0.0057$ |
| | DNFS | $0.9685 \pm 0.0010$ | $-2.1120 \pm 6.6258e^{-05}$ | $-1.4743 \pm 0.0068$ | $0.2811 \pm 0.0030$ |

Table 3: Experiment results of probability mass estimation on seven synthetic datasets. We display the negative log-likelihood (NLL) and MMD (in units of $1 \times 10^{-4}$).

| Metric | Method | 2spirals | 8gaussians | circles | moons | pinwheel | swissroll | checkerboard |
|---|---|---|---|---|---|---|---|---|
| NLL↓ | PCD | 20.094 | 19.991 | 20.565 | 19.763 | 19.593 | 20.172 | 21.214 |
| | ALOE+ | 20.062 | 19.984 | 20.570 | 19.743 | 19.576 | 20.170 | 21.142 |
| | ED-Bern | 20.039 | 19.992 | 20.601 | 19.710 | 19.568 | 20.084 | 20.679 |
| | EB-GFN | 20.050 | 19.982 | 20.546 | 19.732 | 19.554 | 20.146 | 20.696 |
| | EB-DNFS | 20.118 | 19.990 | 20.517 | 19.789 | 19.566 | 20.145 | 20.682 |
| MMD↓ | PCD | 2.160 | 0.954 | 0.188 | 0.962 | 0.505 | 1.382 | 2.831 |
| | ALOE+ | 0.149 | 0.078 | 0.636 | 0.516 | 1.746 | 0.718 | 12.138 |
| | ED-Bern | 0.120 | 0.014 | 0.137 | 0.088 | 0.046 | 0.045 | 1.541 |
| | EB-GFN | 0.583 | 0.531 | 0.305 | 0.121 | 0.492 | 0.274 | 1.206 |
| | EB-DNFS | 0.603 | 0.070 | 0.527 | 0.223 | 0.524 | 0.388 | 0.716 |

rate matrix. Furthermore, Figure 15 compares the loss values between DNFS and LEAPS, showing that our approach again outperforms the baseline. This demonstrates the combined effectiveness of our coordinate descent learning algorithm and transformer-based architecture.

**Quantatitive Results.** Recall Equation (11), where the Ising model is defined as

$$p(x) \propto \exp(-E(x)) \triangleq \exp(\sigma x^T A_D x), \quad x \in \{-1, 1\}^D \tag{53}$$

where $\sigma \in \mathbb{R}$ is the temperature and $A_D$ denote the adjacency matrix of the lattice graph. We evaluate our method by comparing the estimated free energy $\mathcal{F} = -\frac{1}{2\sigma} \log Z$, internal energy $\mathcal{E} = \mathbb{E}_p[E(x)]$, and entropy $\mathcal{S} = 2\sigma(\mathcal{E} - \mathcal{F})$ with their theoretically optimal values derived in Ferdinand & Fisher (1969). The free energy and internal energy are estimated using Equations (37) and (38), respectively. We compare DNFS with LEAPS under two temperature settings, $\sigma = 0.1$ and $\sigma = 0.22305$, where the latter corresponds to the critical temperature and thus presents a more challenging sampling problem. The result is reported in Table 2, where we use $2,048$ Monte Carlo samples to estimate the values and report the mean and standard deviation averaged over 10 independent runs. The results show that DNFS provides accurate estimates of the free and internal energies, closely matching the theoretical value with low variance. In contrast, LEAPS exhibits significantly lower ESS and larger deviations in other metrics under the critical temperature setting, which may indicate insufficient mode coverage. These comparisons highlight the robustness of DNFS and suggest it is less prone to mode collapse across different temperatures.

### E.2 Training Discrete EBMs

#### E.2.1 Experimental Details

**Probability Mass Estimation.** This experiment follows the setup of Dai et al. (2020). We first sample 2D data points $\hat{x} \triangleq [\hat{x}_1, \hat{x}_2] \in \mathbb{R}^2$ from a continuous distribution $\hat{p}$, and then quantise each point into a 32-dimensional binary vector $x \in \{0, 1\}^{32}$ using Gray code. Formally, the resulting discrete distribution follows $p(x) \propto \hat{p}([\text{GradyToFloat}(x_{1:16}), \text{GradyToFloat}(x_{17:32})])$.

Following Schröder et al. (2024), we parameterise the energy function using a 4-layer MLP with 256 hidden units and Swish activation. The leTF model consists of 3 bidirectional causal attention layers, each with 4 heads and 128 hidden units. The probability path is defined by a linear schedule over 64

Table 4: Experimental Results of EBM and DNFS on probability mass estimation: Rows labelled 'EBM' represent metrics evaluated using the trained EBM model, while rows labelled 'DNFS' represent metrics evaluated using the trained DNFS.

| Metric | Method | 2spirals | 8gaussians | circles | moons | pinwheel | swissroll | checkerboard |
|--------|--------|----------|------------|---------|-------|----------|-----------|--------------|
| NLL↓ | EBM | 20.118 | 19.990 | 20.517 | 19.789 | 19.566 | 20.145 | 20.682 |
| | DNFS | 20.947 | 20.948 | 21.043 | 20.908 | 21.011 | 20.899 | 21.106 |
| MMD↓ | EBM | 2.553 | 1.429 | 0.897 | 2.808 | 1.733 | 0.731 | 6.168 |
| | DNFS | 0.603 | 0.070 | 0.527 | 0.223 | 0.524 | 0.388 | 0.716 |

Figure 16: Additional qualitative results in training discrete EBMs. We visualise the training data, learned energy landscape, and the synthesised samples of DNFS.

time steps, starting from a uniform initial distribution. Both the EBM and DNFS are trained using the AdamW optimiser with a learning rate of 0.0001 and a batch size of 128. Notably, each update step of the EBM is performed after every 10 update steps of DNFS. To ensure numerical stability, the log-ratio term is clipped at a maximum value of 5.

After training, we quantitatively evaluate all methods using negative log-likelihood (NLL) and maximum mean discrepancy (MMD), as reported in Table 3. Specifically, the NLL is computed using the trained EBM on 4,000 samples drawn from the data distribution, with the normalization constant estimated via importance sampling using 1,000,000 samples from a variational Bernoulli distribution with $p = 0.5$. For the MMD metric, we follow the protocol in Zhang et al. (2022a), employing an exponential Hamming kernel with a bandwidth of 0.1. All reported results are averaged over 10 independent runs, where each run uses 4,000 samples generated by the trained DNFS.

**Training Ising Models.** Following Grathwohl et al. (2021); Zhang et al. (2022b), we train a learnable adjacency matrix $J_\phi$ to approximate the true matrix $J$ in the Ising model. To construct the dataset, we generate 2,000 samples using Gibbs sampling with 1,000,000 steps per instance. The leTF model consists of three bidirectional causal attention layers, each with four heads and 128 hidden units. The probability path is defined by a linear schedule over 64 time steps, starting from a uniform distribution. $J_\phi$ is optimised using the AdamW optimiser with a learning rate of 0.0001 and a batch size of 128. To promote sparsity, we follow Zhang et al. (2022a) and apply $l_1$ regularisation with a coefficient of 0.05. DNFS is trained separately using AdamW with a learning rate of 0.001 and the same batch size. To ensure numerical stability, the log-ratio term is clipped at a maximum value of 5. We train both models iteratively, performing one update step for $J_\phi$ for every ten update steps of DNFS.

### E.2.2 Additional Results

**Additional Results of Probability Mass Estimation.** We compare DNFS to various baselines, including PCD (Tieleman, 2008), ALOE+ (Dai et al., 2020), ED-Bern (Schröder et al., 2024), and EB-GFN (Zhang et al., 2022a). As shown in Table 3, DNFS outperforms both Persistent Contrastive Divergence (PCD), which relies on conventional MCMC methods, and the variational approach ALOE+, demonstrating the effectiveness of our method. Additional qualitative results are provided in Figure 16, where DNFS consistently produces accurate energy landscapes and high-quality samples that closely resemble the training data. Furthermore, Figure 17 illustrates the sampling trajectory of DNFS alongside marginal samples from long-run Gibbs sampling. The comparison shows that DNFS produces samples that closely resemble those from Gibbs, demonstrating its ability to approximate the target distribution with high fidelity.

Notably, since the EBM and the sampler are trained jointly, we can evaluate the negative log-likelihood (NLL) of the data using the trained CTMC, as described in Appendix D.1, and assess sample quality via the maximum mean discrepancy (MMD) using samples generated by the trained sampler. In Table 4, we report results under four evaluation settings: i) NLL (EBM): using the trained EBM with importance sampling; ii) NLL (DNFS): using the trained CTMC following the method in Appendix D.1; iii) MMD (EBM): samples drawn via Gibbs sampling from the EBM; and iv) MMD (DNFS): samples generated by DNFS. It can be seen that samples generated by DNFS achieved lower MMD compared to those generated by Gibbs sampling, demonstrating the superiority of the learned sampler in capturing the target distribution and producing higher-fidelity samples. However, we observe that DNFS yields less accurate likelihood estimates compared to importance sampling performed with the trained energy function. This is likely because DNFS is trained to satisfy the Kolmogorov equation rather than explicitly optimising the evidence lower bound, as is common in other discrete diffusion models for generative modelling (Shi et al., 2024; Sahoo et al., 2024). Moreover, it is noteworthy that the sample quality and likelihood are not necessarily consistent (Theis et al., 2016, Section 3.2), and DNFS does not directly optimise the likelihood. Thus, the performance of DNFS on NLL is not guaranteed.

**Additional Results of Training Ising Models.** We further provide a quantitative comparison against baselines for training Ising models. Following Zhang et al. (2022a); Schröder et al. (2024), we evaluate on $D = 10 \times 10$ grids with $\sigma = 0.1, 0.2, \ldots, 0.5$ and $D = 9 \times 9$ grids with $\sigma = -0.1, -0.2$. Performance is measured by the negative log-RMSE between the estimated $J_\phi$ and the true adjacency matrix $J$. As shown in Table 5, while our method underperforms ED-GFN, which is also a neural sampler, it achieves results comparable to Gibbs, GwG, and ED-Bern in most settings, demonstrating its ability to uncover the underlying structure in the data.

Table 5: Mean negative log-RMSE (higher is better) between the learned connectivity matrix $J_\phi$ and the true matrix $J$ for different values of $D$ and $\sigma$.

| Method \ $\sigma$ | $D = 10^2$ | | | | | $D = 9^2$ | |
|---|---|---|---|---|---|---|---|
| | 0.1 | 0.2 | 0.3 | 0.4 | 0.5 | −0.1 | −0.2 |
| Gibbs | 4.8 | 4.7 | 3.4 | 2.6 | 2.3 | 4.8 | 4.7 |
| GwG | 4.8 | 4.7 | 3.4 | 2.6 | 2.3 | 4.8 | 4.7 |
| ED-Bern | 5.1 | 4.0 | 2.9 | 2.6 | 2.3 | 5.1 | 4.3 |
| EB-GFN | 6.1 | 5.1 | 3.3 | 2.6 | 2.3 | 5.7 | 5.1 |
| DNFS | 4.6 | 3.9 | 3.1 | 2.6 | 2.3 | 4.6 | 3.9 |

### E.3 Solving Combinatorial Optimisation Problems

#### E.3.1 Experimental Details

This experiment follows the setup in (Zhang et al., 2023a), where we train an amortised combinatorial solver using DNFS on 1,000 training graphs and evaluate it by reporting the average solution size over 100 test graphs. To be specific, we use Erdős–Rényi (ER) (Erdos, 1961) and Barabási–Albert (BA) (Barabási & Albert, 1999) random graphs to benchmark the MIS and MCut problems, respectively. Due to scalability limitations of our current method, we restrict our evaluation to small graphs with 16 to 75 vertices, leaving the exploration of more complex graphs for future work.

We parameterise the rate matrix using the proposed locally equivariant GraphFormer (leGF), which consists of 5 bidirectional causal attention layers, each with 4 heads and 256 hidden units. Training is performed using the AdamW optimiser with a learning rate of 0.0001 and a batch size of 256. The log-ratio term is clipped to a maximum value of 5 to ensure stability. More importantly, we find that the temperature $T$ plays a crucial role in performance. Fixed temperature values generally lead to suboptimal results. Therefore, we adopt a temperature annealing strategy: starting from an inverse temperature of 0.1 and gradually increasing it to a final value of 5. A more comprehensive annealing strategy, such as adaptive schedules based on loss plateaus, may further improve performance by better aligning the sampling dynamics with the learning process. Exploring such adaptive annealing schemes is a promising direction for future work.

#### E.3.2 Additional Results

**Additional Results on Maximum Cut.** We further evaluate DNFS on the maximum cut problem. As shown in Table 6, the trained DNFS significantly outperforms its untrained version, underscoring the effectiveness of our approach. While DNFS slightly lags behind the baselines DMALA and

Table 6: Maximum cut experimental results. We report the absolute performance, approximation ratio (relative to GUROBI), and inference time.

| METHOD | BA16-20 | | | BA32-40 | | | BA64-75 | | |
|---|---|---|---|---|---|---|---|---|---|
| | SIZE ↑ | DROP ↓ | TIME ↓ | SIZE ↑ | DROP ↓ | TIME ↓ | SIZE ↑ | DROP ↓ | TIME ↓ |
| GUROBI | 40.85 | 0.00% | 0:02 | 93.67 | 0.00% | 0:05 | 194.08 | 0.00% | 0:14 |
| RANDOM | 25.70 | 37.1% | 0:03 | 46.19 | 50.7% | 0:05 | 81.19 | 58.2% | 0:08 |
| DMALA | 40.32 | 1.30% | 0:04 | 93.47 | 0.21% | 0:06 | 192.33 | 0.90% | 0:07 |
| GFLOWNET | 39.93 | 2.25% | 0:02 | 90.65 | 3.22% | 0:04 | 186.60 | 3.85% | 0:07 |
| DNFS | 39.60 | 3.06% | 0:03 | 88.64 | 5.37% | 0:05 | 181.75 | 6.35% | 0:08 |
| DNFS+DMALA | 40.76 | 0.22% | 0:08 | 93.63 | 0.01% | 0:12 | 192.30 | 0.92% | 0:17 |

Table 8: Results for the maximum independent set problem on RB32-40 graphs with varying parameter $p$. Reported values include the solution size (larger is better) and the percentage drop in performance relative to GUROBI (lower is better), indicated in brackets.

| $p =$ | 0.1 | 0.3 | 0.5 | 0.7 | 0.9 | 1.0 |
|---|---|---|---|---|---|---|
| GUROBI | 8.52(0.00%) | 8.24(0.00%) | 7.06(0.00%) | 7.89(0.00%) | 8.05(0.00%) | 8.63(0.00%) |
| Random | 3.38(60.3%) | 4.59(44.2%) | 4.25(39.8%) | 5.54(29.7%) | 6.29(21.8%) | 6.93(19.6%) |
| DMALA | 8.50(0.23%) | 8.16(0.97%) | 7.01(0.70%) | 7.84(0.63%) | 8.04(0.12%) | 8.63(0.00%) |
| GflowNet | 8.20(3.75%) | 7.93(3.76%) | 6.83(3.25%) | 7.78(1.39%) | 8.03(0.62%) | 8.63(0.00%) |
| DNFS | 8.00(6.10%) | 7.65(7.16%) | 6.67(5.52%) | 7.66(2.91%) | 7.97(0.99%) | 8.63(0.00%) |
| DNFS+DMALA | 8.51(0.11%) | 8.19(0.60%) | 7.02(0.56%) | 7.85(0.50%) | 8.04(0.12%) | 8.63(0.00%) |

GFlowNet, its MCMC-refined variant achieves the best overall performance, closely approaching the oracle solution provided by Gurobi.

**MCMC Refined DNSF.** As previously discussed, a key advantage of DNFS is its ability to incorporate additional MCMC steps to refine the sampling trajectory, thanks to the known marginal distribution $p_t$. To validate the effectiveness of this MCMC-refined sampling, we conduct an experiment on the MIS problem using the ER16-20 dataset. In this experiment, we compare two methods: DMALA (Zhang et al., 2022b) and

Table 7: Comparison between DNFS and its DLAMA-refined version by solving MIS on the ER16-20 dataset.

| STEPS | DLAMA | | DNFS+DLAMA | |
|---|---|---|---|---|
| | SIZE ↑ | TIME (S) ↓ | SIZE ↑ | TIME (S) ↓ |
| 1 | 8.33 | 1.75 | 8.37 | 4.53 |
| 2 | 8.53 | 3.19 | 8.63 | 5.89 |
| 3 | 8.71 | 4.70 | 8.75 | 7.62 |
| 4 | 8.77 | 6.12 | 8.84 | 8.85 |
| 5 | 8.80 | 7.74 | 8.91 | 10.47 |

DNFS combined with DLAMA, both sampling from the same interpolated distribution $p_t \propto p_0^{1-t} p_1^t$. For reference, the average solution size obtained by DNFS without DLAMA refinement is 8.28. As shown in Table 7, applying DLAMA refinement significantly boosts performance, with improvements increasing as more refinement steps are added. More importantly, integrating the proposed neural sampler (DNFS) with the MCMC method (i.e., DNFS + DLAMA) outperforms the standalone MCMC baseline (i.e., DMALA), demonstrating that the learned sampler provides a strong initialisation that guides the refinement process toward better solutions. This result confirms the synergy between neural samplers and MCMC refinement in solving challenging combinatorial problems.

**Benchmarking on the RB Graphs.** Following Sanokowski et al. (2024), we further evaluate our method on the Maximum Independent Set problem using the RB32-40 graphs, varying the parameter, which controls the problem's difficulty. Specifically, higher $p$ values of yield easier instances, while lower values result in more challenging graphs. For each setting, we report the solution size, along with the percentage performance drop relative to GUROBI. As shown in Table 8, the results are consistent with the observation in Table 1. On easier instances (i.e., higher values of $p$), our method performs competitively, approaching the performance of the oracle solver GUROBI. On more challenging instances, however, DNFS exhibits a performance gap relative to GFlowNet. This gap arises because GFlowNet restricts its sampling trajectories to the feasible solution, effectively narrowing the exploration space. Introducing such an inductive bias into DNFS represents a promising direction for future work. Nevertheless, despite DNFS underperforming GFlowNet in its pure form, it offers a distinct advantage: the intermediate target distribution is explicitly known. This property enables integration with MCMC-based refinement methods (e.g., DNFS+DMALA), which significantly improves performance.

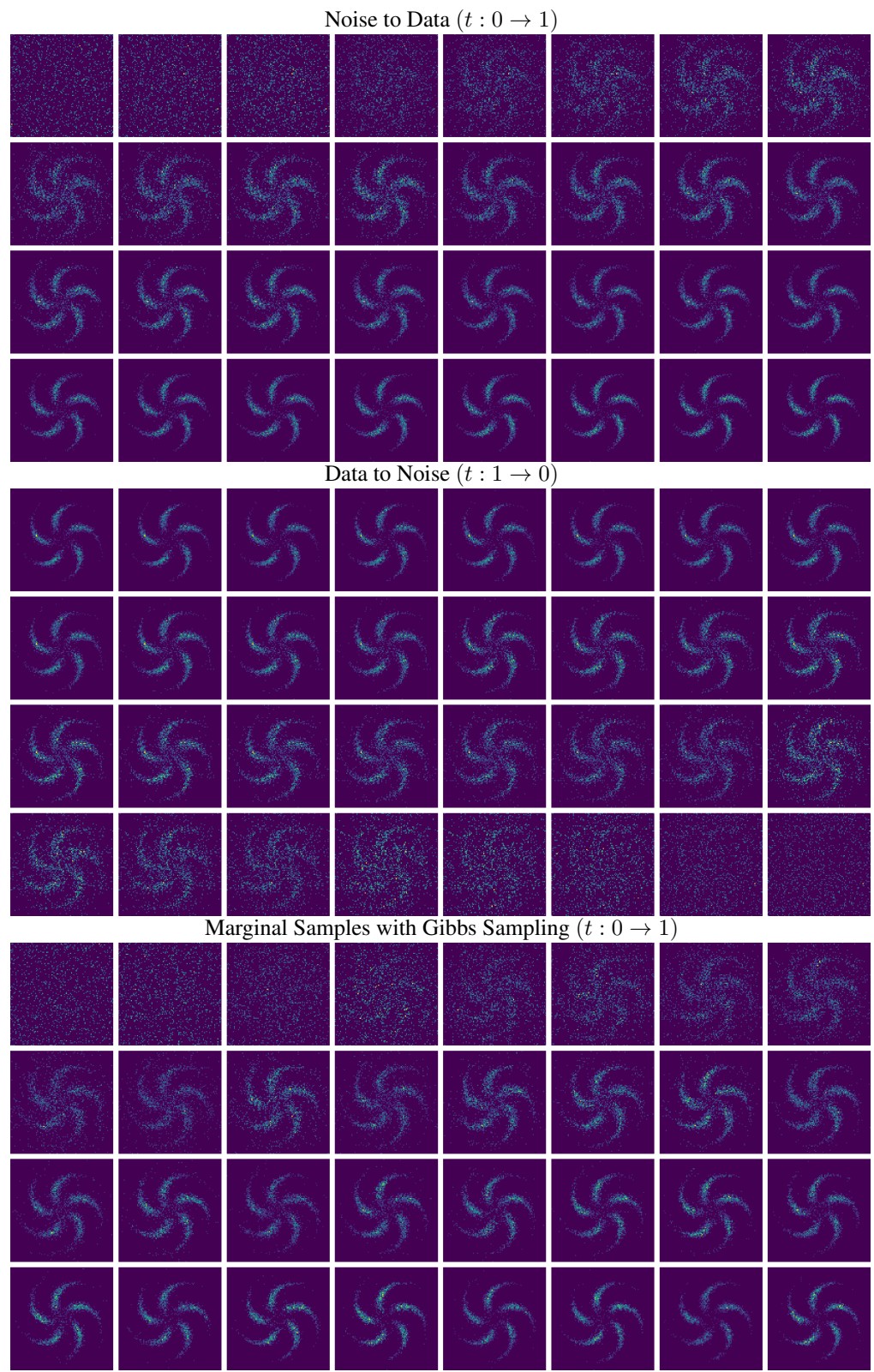

Figure 17: The sampling trajectory of DNFS in discrete EBM training. Top: noise to data trajectory; Middle: data to noise trajectory; Bottom: marginal samples with Gibbs sampling.

