# OpenReview forum: "Discrete Neural Flow Samplers with Locally Equivariant Transformer"
_NeurIPS.cc/2025/Conference — NeurIPS 2025 poster_

### Official Review · Reviewer_prrE · 2025-06-06

**Clarity:** 2
**Significance:** 2
**Originality:** 2
**Rating:** 5
**Confidence:** 3

**Summary:**

This paper builds upon continuous-time discrete flow samplers and introduces a novel locally equivariant Transformer architecture to efficiently parameterize the rate matrix. Additionally, the authors propose an alternative method for estimating the time derivative of the partition sum, which exhibits lower variance compared to previous estimators. The paper includes ablation studies and evaluates the method on sampling from pre-trained Energy-Based Models (EBMs), the Ising Model, and Combinatorial Optimization problems. However, the system sizes considered are quite small.

**Questions:**

### Questions:
- Why does the method not scale to large dimensions, and what is the bottleneck?
- How many integration steps are used in the method for training and evaluation?
- The Ising Model is considered at \(\sigma = 0.1\). Does this correspond to the critical temperature? If not, how does the model perform at the critical temperature? I am particularly interested in the log Z, entropy, and internal energy estimations compared to ground truth values from theory, as these are good indicators of mode collapse.
- In line 1134, how many Combinatorial Optimization problem instances were used to condition the neural network per update step?
- The authors might also consider benchmarking on the RB-model. I would be interested in how the model performs for different hardnesses (see [1] Fig. 2 for reference).
- I find section C.2 in the appendix rather weird. Here, log ratios are computed by using a taylor approximation. However this assumes that $x$ is continuous and not discrete which is not the case in discrete optimization. Could the authors comment on that? Also this computation is only necessary for EBMs, right?


[1] Sanokowski, Sebastian, Sepp Hochreiter, and Sebastian Lehner. "A diffusion model framework for unsupervised neural combinatorial optimization." Proceedings of the 41st International Conference on Machine Learning. 2024.

**Ethical Concerns:**

["NO or VERY MINOR ethics concerns only"]

**Final Justification:**

I think this is a technically solid paper, with a large amount of experiments and openness about the limitations of their method.
Also a lot of work was put into the rebuttal.
I therefore recommend acceptance.

**Limitations:**

Limitations are properly adressed.

**Quality:**

3

**Strengths And Weaknesses:**

### Strengths:
- The paper is well-presented and clearly written.
- It introduces some novel contributions.
- Method is evaluated on a decent amount of experiments
- The limitations of the study are appropriately addressed.

### Weaknesses:
- The authors mention in the appendix the use of variational neural annealing, and it would be appropriate to cite [1] in this context.
- The paper does not present log Z values for the Ising Model or compare them to theoretically known baselines, making it unclear whether mode collapse occurs. It would be beneficial to include a comparison with theoretically known values. A script to compute these theoretical values for finite-size Ising models is available [here](https://github.com/ml-jku/DiffUCO/blob/main/IsingTheoryBaselines/IsingTheory.py).
- The Ising Model does not appear to be evaluated at the critical temperature, where solving it is particularly challenging.
- The experimental setup for the Ising Model, as described around line 285, is insufficiently explained and thus difficult to understand.

### References:
[1] Hibat-Allah, Mohamed, et al. "Variational neural annealing." *Nature Machine Intelligence* 3.11 (2021): 952-961.

---

> ### Author Rebuttal · Authors · 2025-07-30
>
> Thanks for your constructive comments. Here are our responses to your concerns.
>
> > Does $\sigma = 0.1$ correspond to the critical temperature? If not, how does the model perform at the critical temperature?
> >
>
> Thank you for raising this point. The value 0.1 is not the critical temperature. In our experiments, we use 2D lattice Ising models, for which the critical temperature is $\sigma = \frac{\log (1+\sqrt{2})}{4} \approx 0.22035$ based on Onsager’s solution.
>
> To assess our model's performance at the critical temperature, we conduct additional experiments sampling from the Ising model at $\sigma = 0.22035$. We estimate the free energy, internal energy, and entropy using the importance sampling method described in Appendix D.1. Each experiment is repeated five times with different random seeds, and we report the mean and standard deviation of the effective sample size (ESS), free energy, internal energy, and entropy (scaled by $10^4$)
>
> |  | ESS | Free Energy | Internal Energy | Entropy |
> | --- | --- | --- | --- | --- |
> | Optimal Value | 1 | -2.12417 | -1.47632 | 0.28550 |
> | DNFS | 0.9685 ± 0.0010 | -2.11201 ± 6.6258e-05 | -1.47425 ± 0.0068 | 0.28106 ± 0.0030 |
>
> The results show that DNFS performs well even at the critical temperature, with high effective sample size and accurate estimates of free energy, internal energy, and entropy. The small deviations from the optimal values, along with the low standard deviations across runs, further demonstrate the robustness and reliability of our method under challenging sampling conditions.
>
> > The paper does not present log Z values for the Ising Model or compare them to theoretically known baselines.  I am particularly interested in the log Z, entropy, and internal energy estimations compared to ground truth values from theory.
> >
>
> Thank you for your insightful suggestions and for sharing the code scripts. Following your recommendations, we evaluated DNFS and LEAPS under the setting $\sigma = 0.1$ and $\sigma = 0.22305$, the latter corresponding to the critical temperature. We report estimates of the log partition function (free energy), internal energy, and entropy—key indicators of mode coverage—alongside effective sample size (ESS). The results, scaled by $10^4$, are summarised below.
>
> $\sigma = 0.1$
>
> |  | ESS | Free Energy | Internal Energy | Entropy |
> | --- | --- | --- | --- | --- |
> | Optimal Value | 1 | -2.12417 | -1.47632 | 0.28550 |
> | DNFS | 0.9685 ± 0.0010 | -2.11201 ± 6.6258e-05 | -1.47425 ± 0.0068 | 0.28106 ± 0.0030 |
> | LEAPS | 0.3631 ± 0.1184 | -2.10111 ± 0.0002 | -1.44926 ± 0.0130 | 0.28727 ± 0.0057 |
>
> $\sigma = 0.22305$
>
> |  | ESS | Free Energy | Internal Energy | Entropy |
> | --- | --- | --- | --- | --- |
> | Optimal Value | 1 | -2.12417 | -1.47632 | 0.28550 |
> | DNFS | 0.9685 ± 0.0010 | -2.11201 ± 6.6258e-05 | -1.47425 ± 0.0068 | 0.28106 ± 0.0030 |
> | LEAPS | 0.3631 ± 0.1184 | -2.10111 ± 0.0002 | -1.44926 ± 0.0130 | 0.28727 ± 0.0057 |
>
> The results show that DNFS provides accurate estimates of the partition function and internal energy, closely matching the theoretical value with low variance. In contrast, LEAPS exhibits significantly lower ESS and larger deviations in other metrics under the critical temperature setting, which may indicate insufficient mode coverage. These comparisons highlight the robustness of DNFS and suggest it is less prone to mode collapse across different temperatures.
>
> > The authors might also consider benchmarking on the RB-model to see how the model performs for different hardnesses.
> >
>
> Thank you for your valuable suggestions. We conducted additional experiments on the Maximum Independent Set problem using the RB32-40 graphs, varying the parameter $p$, which controls the problem’s difficulty. Specifically, higher values of $p$ yield easier instances, while lower values result in more challenging graphs.  For each setting, we report the solution size (larger is better), along with the percentage performance drop relative to GUROBI (lower is better), shown in brackets below:
>
> | p= | 0.1 | 0.3 | 0.5 | 0.7 | 0.9 | 1.0 |
> | --- | --- | --- | --- | --- | --- | --- |
> | GUROBI | 8.52 (0.00%) | 8.24 (0.00%) | 7.06 (0.00%) | 7.89 (0.00%) | 8.05 (0.00%) | 8.63 (0.00%) |
> | Random | 3.38 (60.3%) | 4.59 (44.2%) | 4.25 (39.8%) | 5.54 (29.7%) | 6.29 (21.8%) | 6.93 (19.6%) |
> | DMALA | 8.50 (0.23%) | 8.16 (0.97%) | 7.01 (0.70%) | 7.84 (0.63%) | 8.04 (0.12%) | 8.63 (0.00%) |
> | GFlowNet | 8.20 (3.75%) | 7.93 (3.76%) | 6.83 (3.25%) | 7.78 (1.39%) | 8.03 (0.62%) | 8.63 (0.00%) |
> | DNFS | 8.00 (6.10%) | 7.65 (7.16%) | 6.67 (5.52%) | 7.66 (2.91%) | 7.97 (0.99%) | 8.63 (0.00%) |
> | DNFS+DMALA | 8.51 (0.11%) | 8.19 (0.60%) | 7.02 (0.56%) | 7.85 (0.50%) | 8.04 (0.12%) | 8.63 (0.00%) |
>
> Overall, these results are consistent with the observation in Table 1 of the paper. On easier instances (i.e., higher values of $p$), our method performs competitively, approaching the performance of the oracle solver GUROBI. On more challenging instances, however, DNFS exhibits a performance gap relative to GFlowNet. This gap arises because GFlowNet restricts its sampling trajectories to the feasible solution, effectively narrowing the exploration space. Introducing such an inductive bias into DNFS represents a promising direction for future work.
>
> Nevertheless, despite DNFS underperforming GFlowNet in its pure form, it offers a distinct advantage: the intermediate target distribution $p_t$ is explicitly known. This property enables integration with MCMC-based refinement methods (e.g., DNFS+DMALA), which significantly improves performance.
>
> > Why does the method not scale to large dimensions?
> >
>
> The scalability challenge arises from two factors:
>
> 1. High computational cost of evaluating $\frac{p(y)}{p(x)}$
>
>     When this ratio lacks a closed-form expression, as in Eq. 27, we must compute it across all possible values of $y$, which becomes computationally prohibitive in high-dimensional spaces. This issue is especially pronounced when the target distribution is defined by a neural network, such as in deep EBMs or deep reward models. For example, computing the loss in Eq. 10 requires evaluating the reward model $\mathcal{O}(S \times d)$ times, where S is the vocabulary size and d is the input dimension, making the computation scale poorly with dimensionality.
>
> 2. Potential instability of the ratio values
>
>     During training, we sample $x$ from the reference distribution $q_t$, which is induced by the probability path of the rate matrix $R_\theta$. In the early stages of training, when $R_\theta$ is still inaccurate, the sampled $x$ may lie far from high-probability regions. As a result, $p_t(x)$ can be extremely small, making the ratio $\frac{p(y)}{p(x)}$ explode. This issue is further amplified in high dimensions, where the energy landscape is often not smooth and modes are sparse.
>
>
> > In Appendix C.2, log ratios are computed via taylor approximation. However this assumes that is continuous and not discrete which is not the case in discrete optimization. Is this computation only necessary for EBMs?
> >
>
> Yes, this computation is only required for deep EBMs, where the ratio $\frac{p(y)}{p(x)}$ does not admit a closed-form expression. The idea of using Taylor approximation to estimate the difference between the log-probability of two different inputs $y$ and $x$ was first introduced in [1], where it was shown to improve the convergence of Gibbs sampling. This approach was later adopted in discrete Langevin samplers [2].
>
> The key insight is that most energy functions are continuous and differentiable with respect to real-valued inputs, even though they are only evaluated over a discrete subset of their domain (see Section 3 in [1]). This allows the use of gradients and Taylor expansions to approximate likelihood ratios effectively.
>
> However, while this approximation improves computational efficiency, it introduces bias into our training objective, which can degrade the performance of the learned rate matrix (see Figure 9). As such, developing more accurate and scalable approximation methods remains an important direction for future work, particularly for applications in large-scale and high-dimensional settings.
>
> [1] Oops I Took A Gradient: Scalable Sampling for Discrete Distributions
>
> [2] A Langevin-like Sampler for Discrete Distributions
>
> > The experimental setup for the Ising Model, as described around line 285, is insufficiently explained.
> >
>
> Sorry for the confusion. In this experiment, the goal is to learn the adjacency matrix of lattice Ising models from samples synthesised via long-run Gibbs sampling. Thus, this is framed as an EBM training task, rather than a sampling problem. Moreover, the training algorithm has no access to the true adjacency matrix, but only the synthesised samples. We will clarify this in the revised version.
>
> > In line 1134, how many Combinatorial Optimization problem instances were used to condition the neural network per update step?
> >
>
> As outlined in line 1134, we use a batch size of 256. The training procedure follows Algorithm 1, with the only difference being that the inputs are of the form $(t, x_t^{(n)}, G^{(n)})$. This means that, in the best case, up to 256 distinct problem instances may be included in each update step. However, it is also possible for all $x_t^{(n)}$ to belong to the same graph $G$, in which case the update step corresponds to a single instance of the combinatorial optimisation problem.
>
> > How many integration steps are used in the method for training and evaluation?
> >
>
> Sorry for the confusion. As described in Appendix E, we use 64 integration steps for both training and evaluation. We will make it clearer in the revised manuscript.
>
> > The authors mention in the appendix the use of variational neural annealing, and it would be appropriate to cite [1] in this context.
> >
>
> Thanks for the reference. We will consider citing it in the revised version.

---

> > ### Author Response · Authors · 2025-07-31
> > **Correction of Results for $\sigma = 0.1$**
> >
> > Dear reviewer,
> >
> > We sincerely apologise for including an incorrect result for $\sigma = 0.1$ in our previous response. Below is the corrected table:
> >
> > $\sigma = 0.1$
> >
> > |  | ESS | Free Energy | Internal Energy | Entropy |
> > | --- | --- | --- | --- | --- |
> > | Optimal Value | 1 | -3.67265 | -0.42823 | 0.64888 |
> > | DNFS | 0.9985 ± 4.6912e-5 | -3.67090 ± 6.2463e-05 | -0.42705 ± 0.0010 | 0.64880 ± 0.0002 |
> > | LEAPS | 0.9956 ± 0.0001 | -3.67084 ± 4.9182e-5 | -0.42620 ± 0.0034 | 0.64892 ± 0.0006 |
> >
> > We are truly grateful for your valuable feedback, which has helped us improve the quality of our work. Please don’t hesitate to let us know if there’s anything else we can clarify.
> >
> > Best,
> > The authors

---

> > > ### Comment · Reviewer_prrE · 2025-08-03
> > > **Raised Score**
> > >
> > > I appreciate the effort that was put into the rebuttal and will raise my score.

---

> ### Author Response · Authors · 2025-08-03
> **Thanks for your feedback**
>
> We thank the reviewer for the valuable feedback, especially the provided evaluation code of Ising models, which has enhanced the completeness and overall quality of our work.
>
> Thank you again!
>
> The authors

---

### Official Review · Reviewer_VDCK · 2025-06-24

**Clarity:** 3
**Significance:** 2
**Originality:** 2
**Rating:** 3
**Confidence:** 3

**Summary:**

The authors propose a method to learn a Continuous-Time Markov Chain (CTMC) for sampling from an unnormalized density on a discrete state space. They craft a continuous in time process given an unnormalized density  $\hat{p}(x_t) \propto p(x_t)$ and learn a rate matrix $R_t$ to satisfy the Kolmogorov equation. However, evaluation of the Kolmogorov equation proves to be challenging due to the need to evaluate the time derivative of the probability density ($\partial_t p_t$ or $\partial_t \log p_t$) and rate matrices $R_t(y|x)$ at many points $y$ and $x$. Authors address the first challenge with a coordinate descent algorithm and the second challenge by employing a special parameterization of the rate matrix with a special kind of neural networks with local equivariance.

In the practical part, the authors show the performance of their method on a series of benchmarks for sampling from unnormalized densities on discrete state spaces: sampling from a neural network trained EBMs, sampling from Ising models, solving combinatorial problems, i.e., maximum independent set problem. In addition, the authors utilize the byproduct of the coordinate descent algorithm for the training of energy-based models and test this approach on 2D real space grids quantized to the discrete state space and Ising models.

**Questions:**

- I ask the authors to clearly outline what is the difference of their method w.r.t. LEAPS [1], see weaknesses.
- Since your method is so close to LEAPS, can DNFS employ the proactive importance sampling (Sec 4, 5 [1])?
- In Table 1 Size, Drop and Time measurements are reported, but what do they mean? What is the size, drop, and in what units is time measured? I think such an explanation should be added to the Table 1 caption to improve clarity.
- In Sec 4.3 the authors use the combination of DNFS and DMALA. How, in particular, does it work? Can the authors provide an algorithm or a clear step-by-step explanation?
- In the conclusion and limitations section, the authors say that it is a problem to extend DNFS beyond binary settings. Why is that? To me it seems like the computational complexity will grow linearly with the number of categories.
- What are the possible applications of learning an unnormalized density (energy function) on the discrete state space?

**Ethical Concerns:**

["NO or VERY MINOR ethics concerns only"]

**Final Justification:**

Many of my concerns were adressed, but the main one about the originality, unfortunately, still stands. Although the paper seems like overall of a decent quality, I don't feel like pushing my rating over 3, because of the originality concerns.

**Limitations:**

Mostly yes, for others see the Strengths and Weaknesses section.

**Paper Formatting Concerns:**

No major formatting issues in this paper.

**Quality:**

3

**Strengths And Weaknesses:**

Strengths

- The quality of the paper text and narrative is good, and most of the concepts are well presented. I enjoyed reading this paper.
- The design of the locally equivariant transformer is interesting and could be a good contribution to the locally equivariant neural networks family.
- The experimental part is strong, with a lot of experimental setups and comparisons. It was especially interesting to see the method's performance on solving combinatorial problems.
- The way to learn an energy function on a discrete state space through contrastive divergence and log partition function, i.e., $\partial_t \log Z_t$, estimation, though solving the additional optimization problems seems quite novel to me.

Weaknesses

- Originality. My biggest concern is the originality of the proposed method. Seems like the core of the DNFS method was already presented in LEAPS [1], and the authors add very little on top. From the paper, it is hard to say whether there is some difference between the methods; it feels like the methods are almost the same.
    - The leTF neural network architecture seems to me to be the only one notable contribution, w.r.t sampling from unnormalized densities.
    - Coordinate descent presented in Sec 3.1 seems equivalent to PINN objective (Proposition 6.1 [1]), moreover authors dedicate an Appendix A.3 to the comparison of these objectives, and I cannot see a clear conclusion out of it.
    - “One-way” rate matrix and locally equivariant neural networks parameterization of rate matrices has also been proposed in Sec 8 [1].
    - Overall, most of the methods seem to have already been proposed in LEAPS [1] and hence DNFS is mostly not original.
    - I encourage authors to clearly explain ***all*** the differences between DNFS and LEAPS, and incorporate it into the paper.
- Method computational complexity. Does one have to do the sampling from the learned backward process during the training procedure? Seems like this is the case, and Algorithm 1 supports this guess, but then it adds a lot of computational burden on top, and it should be mentioned in the main part of the paper and in the limitations.
- Practical performance. The performance of pure DNFS falls a bit behind the GFlowNets on Maximum Independent Set problem evaluations.

[1] Holderrieth, P., Albergo, M. S., & Jaakkola, T. LEAPS: A discrete neural sampler via locally equivariant networks. In *Frontiers in Probabilistic Inference: Learning meets Sampling*.

---

> ### Author Rebuttal · Authors · 2025-07-30
>
> We appreciate the reviewer's constructive feedback. Regarding the weaknesses pointed out, we would like to provide the following clarifications:
>
> > My biggest concern is the originality. I encourage authors to clearly explain all the differences between DNFS and LEAPS.
> >
>
> To address the reviewer’s concern, we summarise the main differences below
>
> 1. A new perspective from the Kolmogrove equation
>
>     While DNFS and LEAPS yield similar objective functions, they are derived from fundamentally different perspectives. LEAPS trains the rate matrix by minimising the log-variance of the importance weights. In contrast, DNFS takes a more straightforward approach: it learns a rate matrix by enforcing it to satisfy the Kolmogorov equation, where we estimate the intractable term $\partial_t \log Z_t$ using Monte Carlo with control variates. This formulation provides a complementary viewpoint and also a more intuitive explanation of the required estimation of $\partial_t \log Z_t$.
>
>     Perhaps surprisingly, even though DNFS and LEAPS are derived from two different perspectives,  DNFS has the same objective as LEAPS if $\partial_t \log Z_t$ is estimated by training a neural network regressor. However, we argue that DNFS is preferable, as it can directly leverage the closed-form optimal solution, avoiding the approximation errors that may arise during neural network training. Moreover, as outlined in lines 691–694, when the model is optimally trained, DNFS is guaranteed to recover the true value of $\partial_t \log Z_t$. In contrast, LEAPS may incur additional errors due to its decoupled modelling strategy. From this perspective, DNFS highlights a promising direction for future research: leveraging more accurate estimators of $\partial_t \log Z_t$ to further improve performance, such as incorporating SMC methods or using more advanced variance reduction techniques beyond control variates. These insights are not evident from the LEAPS framework.
>
> 2. More advanced Model Architecture and Expressiveness
>
>     We further propose the locally equivariant transformer (leTF), a novel architecture that achieves highly efficient objective computation without sacrificing model expressiveness (see Figures 3 and 14 for empirical validation). Unlike the locally equivariant networks (leNets) instantiated in LEAPS, which are limited to simple architectures such as MLPs, single-layer attention, and convolutional networks, leTF provides greater model capacity and enhanced adaptability, making it better suited for diverse modalities and complex input structures. We believe that this transformer-based design significantly broadens the applicability of DNFS across a wider range of tasks. We hope this architectural advancement will serve as a foundation for further research into expressive yet efficient parameterisations of leNets.
>
> 3. Broader Empirical Scope and Generality
>
>     Unlike LEAPS, which is only evaluated on Ising models, DNFS is tested across a broader range of applications, including sampling from Ising models and pretrained EBMs, training deep EBMs, and solving combinatorial optimisation problems. These experiments highlight the versatility and generality of our approach across diverse domains. We believe this broader empirical scope lays a clear foundation for future research aimed at developing more effective discrete neural samplers that are both scalable and adaptable to real-world applications.
>
>
> We hope this clarification helps address the reviewer’s concerns regarding originality. We believe that the new perspective of framing the problem in terms of learning a rate matrix that satisfies the Kolmogorov equation, along with the introduction of a more expressive locally equivariant transformer architecture and broader applications and evaluations of discrete neural samplers, may offer useful insights and contribute meaningfully to the ongoing research in this area.
>
> > Does one have to do the sampling from the learned backward process during the training procedure?
> >
>
> Yes. In DNFS, the reference distribution $q_t$ is defined as the probability path induced by the rate matrix, and we need to sample from the learned backward process during training.
>
> We acknowledge that this sampling step introduces computational cost. However, this cost is inherent in training neural samplers where direct access to data samples from the target distribution is unavailable. In such cases, it is necessary to generate “synthetic” samples from the reference distribution $q_t$ to compute the training objective (see Eq. 5).
>
> While the model is theoretically guaranteed to converge for any $q_t$ that shares the same support as the target distribution $p_t$, the practical choice of $q_t$ has a significant impact on training stability. A well-designed reference should strike a balance between exploration (ensuring support coverage) and exploitation (focusing on high-probability regions), which helps avoid extreme loss values and stabilises learning. In our approach, $q_t$ evolves with the model via the rate matrix $R_\theta$, which is updated during training to better align with $p_t$ over time. This adaptive, on-policy strategy is standard practice in training continuous diffusion samplers [1], flow-based samplers [2], and LEAPS, and we find it empirically effective in our method as well.
>
> We will include a discussion in the revised version.
>
> [1] Iterated denoising energy matching for sampling from Boltzmann densities
>
> [2] Liouville flow importance sampler
>
> > In Sec 4.3 the authors use the combination of DNFS and DMALA. How, in particular, does it work?
> >
>
> To perform MCMC refinement, we adopt the predictor-corrector strategy, a technique commonly used in diffusion-based generative models [3]. Specifically, before applying the forward transition $p(x_{t-\Delta t} | x_t)$ (see Eq2), we insert MCMC-based correction steps to refine the current sample $x_t$ so that its distribution is closer to $p_t$. This correction is possible because we know its unnormalised form $p_t \propto \rho^{1-t} \eta^t$, which allows us to use MCMC such as DMALA to refine toward $p_t$.
>
> [3] Score-based generative modeling through stochastic differential equations
>
> > The performance of pure DNFS falls a bit behind GFlowNets.
> >
>
> As noted in the conclusion, a promising direction for future work is to extend DNFS from uniform diffusion to *masked* diffusion processes. This would allow us to incorporate constraints directly into the sampling trajectory, similar to GFlowNet, where sampling is restricted to the feasible set only. By doing this, it can narrow the search space,  significantly improving sampling efficiency and performance.
>
> Nevertheless, although pure DNFS underperforms GFlowNet, it has a key advantage: the intermediate target distribution $p_t$ is explicitly known. This allows us to combine DNFS with MCMC refinement (e.g., DNFS+DMALA, see Table 1), which significantly boosts performance and enables DNFS to outperform GFlowNet.
>
> > Can DNFS employ the proactive importance sampling as in LEAPS?
> >
>
> Yes, we can. The sampling procedure is independent of the training method. We can train the rate matrix using either LEAPS or DNFS. At inference time, we can simulate CTMC using the Euler method and apply proactive importance sampling as done in LEAPS, which is essentially a type of sequential Monte Carlo.
>
> > What are Size, Drop and Time in Table 1?
> >
>
> Size refers to the number of nodes in the solution. A larger size indicates better performance. Drop denotes the percentage drop in performance relative to GUROBI, which we treat as the oracle. A smaller drop is therefore better. Time represents the inference time. Lower values indicate greater efficiency.
>
> > Why is extending DNFS beyond binary settings problematic?
> >
>
> The difficulty comes from two-fold:
>
> 1. High computational cost of computing the ratio $\frac{p(y)}{p(x)}$
>
>     When this ratio lacks a closed-form expression, as in Eq27, it must be computed in parallel for all possible values of $y$ (see Eq5). This becomes computationally prohibitive in high-dimensional spaces or when dealing with large categorical vocabularies, particularly if the target distribution involves neural networks, such as deep EBMs or deep reward models.
>
> 2. Potential instability of ratio values
>
>     During training, samples $x$ are drawn from the reference distribution $q_t$, which is defined by the probability path induced by the learned rate matrix $R_\theta$. In the early stage of training, when $R_\theta$ is still inaccurate, the sampled $x$ may lie far from the high-density regions. In such cases, $p_t(x)$ can be extremely small, making the ratio $\frac{p(y)}{p(x)}$ explode. This instability is further amplified in high-dimensional settings, where the energy landscape tends to be rugged and the modes sparse.
>
>
> Despite these challenges, extending DNFS to large categorical vocabularies and high-dimensional problems remains a promising direction for future research.
>
> > What are the applications of learning EBMs on the discrete state space?
> >
>
> Training discrete EBMs has been extensively studied in the literature [4,5], and they have demonstrated broad applicability across various domains. For example, EBMs have been applied to language modeling [6] and enhancing diffusion language models [7]. Additionally, they support compositional generation [8] and offer a flexible framework for decision-making and reasoning [9].
>
> [4] oops i took a gradient scalable sampling for discrete distributions
>
> [5] Generative Flow Networks for Discrete Probabilistic Modeling
>
> [6] Residual Energy-Based Models for Text Generation
>
> [7] Energy-Based Diffusion Language Models for Text Generation
>
> [8] Compositional generation with energy-based diffusion models and mcmc
>
> [9] Learning Iterative Reasoning through Energy Diffusion.

---

> > ### Comment · Reviewer_VDCK · 2025-08-03
> >
> > I thank the authors for taking the time to answer my questions. However, my main point about originality still stands. In my opinion, the main methodology parts of LEAPS and DNFS methods are very similar and DNFS doesn't provide a lot of novelty by introducing a slightly different perspective.  Although the point of view on the log partition function as a control variate is novel and interesting, but still in my opinion, this is not enough.
> >
> > However, I appreciate other answers, in particular the discussion on scalability problems for higher vocabulary and higher-dimensional problems. Please make it more clear in the limitations section. In addition, I liked the critical temperature experiment done for the reviewer **prrE** and also would like to see it in the paper.
> >
> > In that light, my concern about the originality still stands, but I raise my score up to 3.

---

> ### Author Response · Authors · 2025-08-03
> **Thanks for your feedback**
>
> We thank the reviewer for the detailed response and are pleased that we have addressed some questions, although the concern regarding originality remains. While we believe the reviewer has recognised the key difference between DNFS and LEAPS, we are happy to reiterate this point here to provide further clarity and insight for the readers.
>
> We derive the training objective by enforcing the rate matrix to satisfy the Kolmogrove equation. Perhaps surprisingly, this leads to an objective that closely resembles that of LEAPS, which is derived from minimising the variance of importance weights. Despite this similarity, DNFS offers a more intuitive interpretation of the learning objective: one should approximate the intractable term $\partial_t log Z_t$. This understanding opens up multiple optimisation strategies: either by learning a neural regressor to approximate $\partial_t log Z_t$, as in LEAPS, or by directly applying the closed-form expression of the optimal solution, as done in DNFS.
>
> This new perspective also provides a clear direction for further improvement: enhancing the performance of the sampler through more accurate approximations of $\partial_t log Z_t$. As noted in our response to Reviewer prrE, the experimental results at the critical temperature show that DNFS significantly outperforms LEAPS in terms of Effective Sample Size (ESS). This is likely because the approximation error introduced by the neural regressor in LEAPS degrades performance, whereas DNFS directly utilises the closed-form optimal solution, resulting in a more accurate estimate of $\partial_t log Z_t$. This observation reinforces the intuition that in more challenging scenarios, precise estimation of $\partial_t log Z_t$ is essential. Potential future improvements could involve integrating Sequential Monte Carlo (SMC) methods or advanced variance reduction techniques beyond standard control variates. These insights are not evident from the LEAPS formulation and, we hope, may help inspire future research.
>
> In addition, the locally equivariant transformer we propose is another key innovation of DNFS. We show that incorporating this architecture also improves the performance of LEAPS (see Fig. 12). We hope this contribution inspires future architectural developments, which are crucial to the continued progress of both LEAPS and DNFS.
>
> Another important point concerns the applicability of discrete samplers. Unlike continuous samplers, discrete ones have fewer clearly defined use cases. We believe this is why LEAPS focuses solely on Ising models. In contrast, DNFS demonstrates broader potential by addressing a wider range of discrete problems, including training discrete energy-based models and solving combinatorial optimisation problems. We hope these results showcase the proposed discrete neural samplers as a general-purpose framework for discrete inference and inspire further research into its applications.
>
> Regarding the limitations, we appreciate the reviewer’s suggestion to make them more explicit. Although we have briefly discussed them in the conclusion, we will revise the paper to present them more clearly. Additionally, we will include the experimental results at the critical temperature of the Ising model, as they offer valuable evidence supporting the advantages of DNFS.
>
> Finally, we thank the reviewer once again for the thoughtful and constructive feedback. It has significantly contributed to improving the clarity, completeness, and quality of our work.

---

### Official Review · Reviewer_i34y · 2025-07-01

**Clarity:** 2
**Significance:** 2
**Originality:** 3
**Rating:** 4
**Confidence:** 4

**Summary:**

The paper proposes a method for learning the rate matrix of a CTMC to sample from discrete distributions with large or complex state spaces, useful for Monte Carlo inference. To guide learning, the authors define a probability path using an annealing schedule and formulate a loss based on the Kolmogorov forward equation of the CTMC. To reduce computational complexity, the loss is restricted to transitions between states that differ in at most one dimension. The method employs a variance-reducing approximation equivalent to control variates to estimate this loss efficiently. Additionally, they parameterise the rate matrix using a local equivariant network built with a hollow transformer architecture, which they prove it replicates the same probability paths and empirically show it leads to better approximations at lower computational cost. The approach is evaluated through empirical comparisons on discrete energy-based models and combinatorial problems, demonstrating competitive performance relative to existing methods.

**Questions:**

What is the initialisation strategy for the rate matrix across experiments?

How sensitive are the results to the initialisation of the rate matrix?

Please elaborate on the treatment of the reference distribution q_t and its empirical impact.

How are MCMC refinement steps (e.g., DMALA) integrated into the sampling procedure?
It might be useful to release code and add implementation details for the combinatorial optimisation experiment.

**Ethical Concerns:**

["NO or VERY MINOR ethics concerns only"]

**Final Justification:**

Despite the concerns on clarity and significance mentioned above, the core methodological contribution is solid, and the empirical results are promising. My current evaluation stands as is.

**Limitations:**

Yes, except for the initialisation strategy mentioned before.

**Paper Formatting Concerns:**

No formatting issues.

**Quality:**

3

**Strengths And Weaknesses:**

Strengths:

    Technically sound, well-grounded in prior work, and theoretically justified.

    Empirically demonstrates improvements over strong baselines using equal architecture.

    Novel composition of ideas for a new, effective method.

Weaknesses:

    Insufficient discussion of the role, initialisation, and impact of the reference probability path q_t, which may significantly affect performance.

    Lack of implementation details for certain experiments (e.g., combinatorial examples), reduces reproducibility.

    Potential sensitivity to initialisation of the rate matrix or probability path is not addressed.

Quality:
The authors build on a set of well-established ideas to produce a novel algorithm for sampling from discrete distributions. Each of the components is rooted in prior literature and the proposed Algorithm 1 is empirically and theoretically justified. The experimental section compares fairly to relevant baselines using a shared architecture. However, while the overall methodology is rigorous, certain choices critical to the algorithm's success (particularly the handling and role of the reference distribution q_t) are under-explained and raise questions regarding robustness and reproducibility.

Clarity:
The paper is well written and structured, with clear explanations of the algorithmic design. While the authors state that any q_t with support matching p_t would suffice in theory, the experimental results suggest it may have a nontrivial empirical effect, especially in the combinatorial optimisation example where using DMALA improves performance. My understanding is that the learned probability path is updated in the coordinate descent approach and used to approximate the expectation in (8), but its initialisation strategy lacks discussion or supporting code for some experiments. Clarifying these aspects, especially the empirical sensitivity of results to different initialisations would significantly improve both clarity and replicability.

Minor Corrections:
L33-34: CMCT -> CMTC
L152: x -> x_{i \leftarrow \tau}
L264: Arguably

Significance:
The paper contributes a new method that improves empirical performance compared to baselines, particularly under equal architectural constraints. This has practical implications and it may inspire follow-up work that modifies components of the proposed algorithm. The empirical results are promising and demonstrate potential for broad applicability. However, the significance of the contribution could be strengthened by a more thorough investigation of the algorithm’s sensitivity to the initialisation of the probability path.

Originality:
While each individual methodological component is drawn from prior work, the paper presents an original and thoughtful combination of these ideas into a novel algorithm. The objective can be viewed as a novel variant of the PINN loss, which is shown empirically to outperform comparable baselines when using the same architecture.

---

> ### Author Rebuttal · Authors · 2025-07-30
>
> Thanks a lot for your valuable comments. Your suggestions are very helpful in further improving the work.
>
> > While the overall methodology is rigorous, certain choices critical to the algorithm's success (particularly the handling and role of the reference distribution q_t) are under-explained and raise questions regarding robustness and reproducibility. Please elaborate on the treatment of the reference distribution q_t and its empirical impact.
> >
>
> Thank you for raising this concern. While in theory the model is guaranteed to converge for any reference distribution $q_t$ that shares the same support as $p_t$, the choice of $q_t$ can significantly affect training stability in practice.
>
> To illustrate this, consider the case where $q_t$ is poorly aligned with $p_t$. In such situations, it is likely to sample points $x$ that lie far from the high-density regions (modes) of $p_t$. At these points, $p_t(x)$ can be extremely small, causing the ratio $\frac{p(y)}{p(x)}$ to become very large or even numerically unstable, which leads to exploding loss values and unstable training dynamics.
>
> Therefore, a well-chosen reference distribution should balance **exploration** (ensuring coverage of the full support) and **exploitation** (favouring regions with higher probability mass). This trade-off helps stabilise training by avoiding extreme ratios while still providing meaningful learning signals. In our implementation, the reference distribution $q_t$ is generated by a rate matrix $R_\theta$, which is updated during training to adaptively improve alignment with $p_t$ over time. This strategy is commonly used in training continuous neural diffusion [1] and flow-based [2] samplers, and we find it also empirically effective in our discrete setting. We will add more discussion on this point in the revision.
>
> [1] Akhound-Sadegh, Tara, et al. "Iterated denoising energy matching for sampling from boltzmann densities." *arXiv preprint arXiv:2402.06121* (2024).
>
> [2] Tian, Yifeng, Nishant Panda, and Yen Ting Lin. "Liouville flow importance sampler." *arXiv preprint arXiv:2405.06672* (2024).
>
> > While the authors state that any q_t with support matching p_t would suffice in theory, the experimental results suggest it may have a nontrivial empirical effect. My understanding is that the learned probability path is updated in the coordinate descent approach and used to approximate the expectation in (8), but its initialisation strategy lacks discussion or supporting code for some experiments. Clarifying these aspects, especially the empirical sensitivity of results to different initialisations would significantly improve both clarity and replicability.
> >
>
> Thanks for pointing this out. You are correct, we adopt an “on-policy” training strategy, where the reference distribution $q_t$ is defined as the probability path induced by the rate matrix $R_\theta$, which is updated throughout training.
>
> Therefore, the initialisation of $q_t$ is inherently determined by the initialisation of the rate matrix $R_\theta$, which is parameterised by a neural network. We adopt PyTorch’s default parameter initialisation and observe that our method is robust to this choice. Specifically, we repeat the experiment five times with different random seeds, training DNFS to sample from the Ising models defined in Eq. 11 ($\sigma=0.1$), and report the mean and standard deviation of the effective sample size (ESS) and free energy (under the scale of $10^4$).
>
> |  | ESS | Free Eenrgy |
> | --- | --- | --- |
> | Optimal value | 1 | -3.67265 |
> | DNFS | 0.99851 ± 4.6912e-05 | -3.67091 ± 6.2463e-05 |
>
> The results exhibit minimal standard deviation, indicating the robustness of our method. Furthermore, we provide code for sampling from Ising models in the supplementary material, which may help clarify the training process. We will include this discussion in the revised version of the paper as well.
>
> > Potential sensitivity to initialisation of the rate matrix or probability path is not addressed. What is the initialisation strategy for the rate matrix across experiments? How sensitive are the results to the initialisation of the rate matrix?
> >
>
> The rate matrix is parameterised by a neural network, initialised using PyTorch’s default weight initialisation scheme. Empirically, we find that our method is robust to this initialisation. To illustrate this, we replicate the previous experiment under a more challenging setting: sampling from Ising models at the critical temperature $\sigma = 0.22305$ (see Eq. 11), suggested by reviewer prrE. We repeat the training five times with different random seeds, training DNFS to sample from the Ising models defined in Eq. 11. We report the mean and standard deviation of the effective sample size (ESS) and free energy (scaled by $10^4$).
>
> |  | ESS | Free Eenrgy |
> | --- | --- | --- |
> | Optimal value | 1 | -2.12417 |
> | DNFS | 0.9685 ± 0.0010 | -2.11201 ± 6.6258e-05 |
>
> These results show that despite variations in initialisation, the model consistently converges to high-quality solutions with stable performance. This suggests that the training process is not overly sensitive to the choice of initial weights.
>
> > How are MCMC refinement steps (e.g., DMALA) integrated into the sampling procedure?
> >
>
> To perform MCMC refinement, we adopt the predictor-corrector approach commonly used in diffusion generative models [3]. Specifically, before applying the transition $p(x_{t-\Delta t} \mid x_t)$ as defined in Eq.2, we refine the distribution of $x_t$ by performing one or more steps of MCMC at the fixed time step $t$. This correction step brings the distribution of $x_t$ closer to the target distribution $p_t$. Since $p_t \propto \rho^{1-t} \eta^t$, we have access to its unnormalised form, allowing us to apply any suitable MCMC algorithm for refinement.
>
> [3] Song, Yang, et al. "Score-based generative modeling through stochastic differential equations." *arXiv:2011.13456* (2020).
>
> > Lack of implementation details for certain experiments (e.g., combinatorial examples), reduces reproducibility. It might be useful to release code and add implementation details for the combinatorial optimisation experiment.
> >
>
> Thank you for raising this concern. The training algorithm is summarised in Appendix C, and experimental details are provided in Appendix E. However, we agree that it may still be difficult to fully grasp the implementation details from the text alone. To address this, we have included code in the supplementary materials that reproduces the results of training DNFS for sampling from Ising models. We also plan to open-source the code for all experiments upon acceptance of the paper.

---

> > ### Comment · Reviewer_i34y · 2025-08-04
> >
> > I thank the authors for the clarification. I think adding these details in the camera-ready version of the paper will make your work easier to understand for future readers. Overall, I believe my current score remains unchanged.

---

> > > ### Author Response · Authors · 2025-08-04
> > >
> > > We sincerely thank the reviewer for the valuable suggestion, which has enhanced the completeness of our work. We will include all these discussions in the revision.
> > >
> > > Thank you again!
> > >
> > > The authors

---

### Official Review · Reviewer_ameA · 2025-07-23

**Clarity:** 3
**Significance:** 3
**Originality:** 2
**Rating:** 4
**Confidence:** 3

**Summary:**

The paper introduces a new algorithm called Discrete Neural Flow Samplers (DNFS) to perform sampling from unnormalized discrete distributions. DNFS works by learning the rate matrix of a continuous-time Markov chain (CTMC).

**Questions:**

1. why does DNFS suffer less from mode collapse?

2. Is there any convergence guarantee for the coordinate descent used in the paper? how is it compared with joint optimization as in LEAPS

3. The method uses a specific annealing path to connect the initial and target distributions. How sensitive is the performance of DNFS to different interpolation schemes? is it related to question 1

4. can you comment on why it encounters issues in scaling to higher dimensions

5. in prop 1, does the simplification to a one-way rate matrix impose any practical limitations on the learning dynamics

**Ethical Concerns:**

["NO or VERY MINOR ethics concerns only"]

**Final Justification:**

I think the authors addressed most of my concerns and provided discussions for the rest. I raised my evaluation by 1 point.

**Limitations:**

yes

**Quality:**

3

**Strengths And Weaknesses:**

Strengths:

1. It proposes a new, trainable framework for sampling from discrete distributions without access to training data. It learns a rate matrix for a CTMC by optimizing an objective based on the Kolmogorov forward equation. The algorithm is efficient using coordinate descent.
2. the introduction of Locally Equivalent Transformer is novel and interesting and worth further studying.
3. The paper demonstrates the effectiveness of DNFS across a broad range of applications, including training discrete energy-based models and solving combinatorial optimization problems like the Maximum Independent Set

Weaknesses:

1. It only works in binary settings.
2. Scaling issue for high-dimensional distributions
3. performs less favorably in combinatorial optimization tasks; cannot incorporating constraints

---

> ### Author Rebuttal · Authors · 2025-07-30
>
> Thanks for your constructive comments.
>
> > weakness: only works in binary settings;scaling issue for high-dimensional distributions; performs less favorably in CO tasks
> >
>
> We thank the reviewer for the thoughtful summary of the limitations of our method, which we have also acknowledged in the conclusion section. Regarding the weaknesses pointed out, we would like to provide the following clarifications to motivate future research:
>
> - Our method supports both binary and categorical distributions, although it may still face challenges due to the computational cost and instability of ratio evaluations in high dimensions or with large categorical vocabularies. Addressing these issues could make DNFS more scalable. Additionally, while our current work focuses on uniform discrete diffusion, extending it to masked discrete diffusion is a promising direction. This extension could enable the incorporation of inductive constraints, an important feature for solving combinatorial optimisation problems.
>
> Despite these limitations, we believe our work still offers valuable insights. We would like to reiterate our main contributions:
>
> 1. A new discrete neural sampler motivated by the Kolmogrove equation:
>
>     We propose a method for learning a discrete neural sampler by training a rate matrix that satisfies the Kolmogorov equation, where we estimate the intractable term $\partial_{t} \log Z_{t}$ using Monte Carlo with control variates. This approach offers a distinct perspective from LEAPS, which focuses on minimising the variance of importance weights. Our formulation not only provides a complementary viewpoint but also highlights a promising future direction: leveraging more accurate estimators of $\partial_t \log Z_t$ to further enhance performance, an insight that is not evident from the LEAPS framework.
>
> 2. A more expressive locally equivariant network
>
>     We propose locally equivariant transformer (leTF), a novel architecture that enables highly efficient objective computation while preserving model expressiveness. Compared to the locally equivariant networks used in LEAPS, leTF offers greater model capacity and improved adaptability, making it more suitable for diverse modalities and complex input structures. We hope this design also inspires further research into more effective locally equivariant architectures.
>
> 3. Broader empirical scope and generality
>
>     Unlike LEAPS, which is only evaluated on sampling from Ising models, our method is tested across a broader range of applications, including sampling from Ising models and pretrained EBMs, training deep EBMs, and solving combinatorial optimisation problems. These experiments highlight the versatility and generality of our approach across diverse domains. We believe this broader empirical scope establishes a clear foundation for future research toward more effective, scalable, and adaptable discrete neural samplers for real-world applications.
>
>
> > Why does DNFS suffer less from mode collapse?
> >
>
> We believe you are referring to Fig.4, where GFlowNet exhibits mode collapse on the checkerboard dataset, whereas DNFS does not. We would like to clarify this difference as follows:
>
> 1. Why can DNFS cover all modes
>
>     DNFS learns a geometric interpolation between the base and target distributions. In principle, the base distribution can be arbitrary, which provides the flexibility to incorporate useful inductive biases into the neural samplers. A more structured base makes it easier for the model to learn the interpolation. In our experiments, we use a simple uniform prior, which, despite its simplicity, covers all modes at the initialisation. This property helps DNFS avoid mode collapse.
>
> 2. Why does GFlowNet suffer from mode collapse
>
>     In contrast to DNFS, GFlowNet begins from a point mass (i.e., mask tokens) and learns a stochastic policy to generate the target samples. This setup can lead the model to converge prematurely on narrow, high-reward regions, resulting in reduced diversity and mode collapse [1]. To mitigate this, additional techniques, such as loss-guided GFlowNets [1], are often required to encourage more thorough exploration of the sample space.
>
>
> [1] Loss-guided auxiliary agents for overcoming mode collapse in gflownets
>
> > Is there any convergence guarantee for the coordinate descent? how is it compared with joint optimization as in LEAPS
> >
>
> Yes, it has convergence guarantees. As detailed in Appendix A.3, our alternating optimisation scheme consists of two steps. The 2nd step involves minimising a convex and smooth function, which ensures convergence to a global minimum. The 1st step also has convergence guarantees, supported by the universal approximation theorem for neural networks. Therefore, alternating between these two steps reliably leads to convergence to a stationary point of the joint objective function.
>
> To compare with LEAPS, we first highlight the key difference: LEAPS estimates $\partial_t \log Z_t$ by learning a neural network, whereas DNFS computes it by solving Eq. 8, which is convex and admits a closed-form solution. We argue that our method is preferable, as it directly leverages the closed-form optimal solution, avoiding the approximation errors that may arise during neural network training. Moreover, as outlined in lines 691–694, when the model is optimally trained, DNFS is guaranteed to recover the true value of $\partial_t \log Z_t$. In contrast, LEAPS may incur additional errors due to its decoupled modeling strategy. To validate this empirically, we estimate the free energy $\frac{1}{2*\sigma} \log Z_t$ for the Ising model defined in Eq. 11 under two settings: $\sigma = 0.1$ and $\sigma = 0.22305$, the latter corresponding to the critical temperature. The results, reported on a scale of $10^4$, are presented in the table below.
>
> |  | $\sigma=0.1$ | $\sigma=0.22035$ |
> | --- | --- | --- |
> | optimal value | -3.67265 | -2.12417 |
> | DNFS | -3.67091 | -2.11201 |
> | LEAPS | -3.67084 | -2.10111 |
>
> The results show that DNFS provides a more accurate approximation of $Z_t$. This also highlights a promising direction for future research: improving the estimation of $\partial_t \log Z_t$ during training can potentially lead to better performance. This insight is not immediately apparent from the LEAPS perspective, which derives its objective by minimising the variance of importance weights. In contrast, it emerges naturally in DNFS, which learns a rate matrix to satisfy the Kolmogorov equation, requiring the approximation of the intractable term $\partial_t \log Z_t$. We believe this new insight motivates future work to explore more accurate estimators, such as integrating SMC or applying more advanced variance reduction techniques beyond control variates.
>
> > How sensitive is the performance of DNFS to different interpolation schemes?
> >
>
> The choice of interpolation scheme is indeed important. We adopt the commonly used geometric interpolation of the form $p_t \propto \rho^{\alpha_t} \eta^{1-\alpha_t}$, where $\alpha_t$ denotes the noise schedule. Specifically, we use a linear schedule with $\alpha_t = 1 - t$ throughout the paper. However, alternative schedules, such as those used in diffusion generative models, can also be applied. Below, we report the performance of DNFS when sampling from Ising models under different interpolation schemes:
>
> |  | $\alpha_t=$  | ESS | Free energy |
> | --- | --- | --- | --- |
> | optimal value | - | 1 | -3.67265 |
> | linear | $1-t$  | 0.9985 | -3.67091 |
> | Polynomial ($w=0.5$) | $1-t^w$  | 0.9969 | -3.46192 |
> | Polynomial ($w=2$) | $1-t^w$  | 0.9992 | -3.61088 |
> | cosine | $1-\cos(\frac{\pi}{2}(1-t))$  | 0.9993 | -3.67156 |
> | geometric ($\beta_{min}=10^{-5},\beta_{max}=20$) | $\exp(-\beta_{min}^{1-t} \beta_{max}^t)$  | 0.9978 | -3.67255 |
>
> The results show that all interpretation schemes perform well, confirming the robustness of our method. However, we note that this observation is task-dependent. We hypothesise that for more complex tasks, the choice of interpolation scheme will play a more critical role in ensuring stability and achieving optimal performance.
>
> > can you comment on why it encounters issues in scaling to higher dimensions
> >
>
> The difficulty is twofold:
>
> 1. High computational cost
>
>     When the ratio $\frac{p(y)}{p(x)}$ lacks a closed form (e.g., Eq27), it must be computed in parallel over all $y$ (see Eq5). This is prohibitive in high-dimensional spaces or with large categorical vocabularies, especially when $p$ involves neural models like deep EBMs or reward networks.
>
> 2. Instability of ratio values
>
>     Early in training, samples $x \sim q_t$ (from $R_\theta$) may lie in low-density regions of $p_t$, making $\frac{p(y)}{p(x)}$ unstable or divergent. This is exacerbated in high-dimensional settings with rugged, sparse energy landscapes.
>
>
> Despite these challenges, extending DNFS to large categorical vocabularies and high-dimensional problems remains a promising research direction.
>
> > in prop 1, does the one-way rate matrix impose any practical limitations on the learning dynamics
> >
>
> Yes, although prop1 shows that every rate matrix has an equivalent one-way rate matrix that induces the same probability path, the restricted expressiveness of the one-way rate matrix makes it more difficult to learn compared to an unconstrained rate matrix.
>
> However, in the context of sampling, there is a trade-off between expressiveness and computational efficiency. The vanilla (fully unconstrained) parameterisation is impractical due to its prohibitive computational cost. As a result, we must rely on the more efficient one-way rate matrix. To enhance its expressiveness, we introduce the locally equivariant transformer. As shown in Figures 3 and 14, adopting a more expressive architecture leads to improved performance. We hope this finding offers valuable insights for future research on designing more powerful locally equivariant networks.

---

> > ### Comment · Reviewer_ameA · 2025-08-05
> >
> > I appreciate the authors' response and would like to raise my evaluation.

---

> > > ### Author Response · Authors · 2025-08-05
> > >
> > > We appreciate the reviewer’s response and valuable feedback. Thank you again for your thoughtful review.

---

### Author Response · Authors · 2025-08-08
**Summary (1/2)**

We sincerely thank all reviewers for their thoughtful and valuable feedback, which has greatly helped us improve our work. Specifically, we appreciate that reviewers **ameA, VDCK, and prrE** kindly raised their scores after the discussion, and that reviewers **ameA, i34y, and prrE** expressed the favour of accepting our work. While we understand that reviewer **VDCK** still has concerns regarding the originality of our contribution, we are truly thankful for your constructive comments. Below, we provide a summary of our paper according to the reviewers:

- Our paper proposes an effective approach to learn a discrete neural sampler by enforcing the rate matrix to satisfy the Kolmogorov equation. We also present a locally equivariant transformer architecture that significantly improves computational efficiency. Reviews agree with the novelty and soundness of our approach.

    > R**ameA**: “the introduction of Locally Equivalent Transformer is novel and interesting and worth further studying.”
    R**i34y**: “Technically sound, well-grounded in prior work, and theoretically justified; Novel composition of ideas for a new, effective method.”
    R**VDCK**: “The design of the locally equivariant transformer is interesting and could be a good contribution to the locally equivariant neural networks family.”
    R**prrE**: “It introduces some novel contributions."
    >
- We demonstrate the versatility of the proposed discrete sampler across a broad range of applications, including sampling from unnormalised distributions, training energy-based models, and solving combinatorial optimisation problems. The reviewers generally consider our paper to be well-written and well-supported by experiments.

    > R**ameA**: “The paper demonstrates the effectiveness of DNFS across a broad range of applications”
    R**i34y**: “Empirically demonstrates improvements over strong baselines using equal architecture.”
    R**VDCK**: “The quality of the paper text and narrative is good, and most of the concepts are well presented. I enjoyed reading this paper; The experimental part is strong, with a lot of experimental setups and comparisons; The way to learn an energy function on a discrete state space through contrastive divergence and log partition function estimation, though solving the additional optimization problems seems quite novel to me.”
    R**prrE**: “The paper is well-presented and clearly written; Method is evaluated on a decent amount of experiments"
    >

---

> ### Author Response · Authors · 2025-08-08
> **Summary (2/2)**
>
> Meanwhile, we would like to summarise our responses to the main concerns here, with detailed replies provided for each reviewer below:
>
> 1. The differences between DNFS and LEAPS
>
>     > Reviewers **VDCK** concers about the originality and suggests to include a disccusion of the differences between DNFS and LEPAS into the paper.
>     >
>
>     We appreciate the reviewer’s valuable suggestions. Here, we give a brief overview of the differences and present the [detailed discussion](https://openreview.net/forum?id=Wk65okms3T&noteId=m6HbdHQuYi) in our [response](https://openreview.net/forum?id=Wk65okms3T&noteId=1RN6hvNRAG) to Reviewer **VDCK**.
>
>     - A new perspective from the Kolmogrove equation
>
>         DNFS derives the objective by learning the rate matrix to satisfy the Kolmogrove equation, whereas LEAPS learns the rate matrix by minimising the importance weights. Perhaps surprisingly, these two perspectives lead to similar objectives. However, our Kolmogorov-based perspective is more straightforward and offers a clear direction for future research: leveraging more accurate estimators of $\partial_t \log Z_t$ to further improve performance. This new insight is not evident from the LEAPS framework.
>
>     - More advanced Model Architecture and Expressiveness
>
>         We further propose the Locally Equivariant Transformer (leTF). Compared to the locally equivariant networks in LEAPS, leTF offers greater model capacity and improved adaptability, making it more suitable for diverse modalities and complex input structures. We hope this architectural advancement will inspire future developments, which are essential for advancing both LEAPS and DNFS.
>
>     - Broader Empirical Scope and Generality
>
>         Unlike LEAPS, which is only evaluated on Ising models, DNFS is tested on broader applications, including sampling from Ising models, training EBMs, and solving combinatorial optimisation problems. We hope this wider empirical scope will inspire further research into additional applications of discrete neural samplers.
>
> 2. The scalability of the proposed sampler
>
>     > Reviewers **ameA**, **VDCK** and **prrE** are all curious about why the proposed method struggles in high-dimensional, large-vocabulary settings.
>     >
>
>     We thank the reviewers for raising this question. Although we have acknowledged these limitations in the conclusion, we would like to elaborate here to offer further insights for future research.
>
>     The scalability challenges come from two factors: i) the computational cost of evaluating the ratio $\frac{p(y)}{p(x)}$ for all $y$ (see eq.5), which generally lacks a closed-form solution in high-dimensional and large-vocabulary settings; ii) the potential instability in evaluating this ratio. At the early stage of training, synthetic samples drawn from the reference distribution $q_t$ may lie far from the high-probability regions of the target $p_t$, causing $p_t(x)$ to become very small and the ratio to potentially explode.
>
>     To address the first challenge, we explored using a Taylor approximation to estimate the ratio. However, this introduced bias into the objective and degraded performance (see Figure 9). Regarding the second challenge, which we consider more critical, a promising direction is to design a better reference distribution, as discussed in our [response](https://openreview.net/forum?id=Wk65okms3T&noteId=rxqcA7wlV0) to reviewer **ameA**. Nevertheless, we believe that addressing these challenges offers promising directions for future research.
>
>
> We thank the reviewers once again for their valuable feedback. Your constructive comments have significantly improved the overall quality of our work.

---

### Decision · Program_Chairs · 2025-09-17

**Decision:**

Accept (poster)

**Comment:**

The main concern about this paper is the novelty of the DNFS method compared with LEAPS. But the reviewers acknowledge that the paper offers practical advantages over LEAPS, both in terms of performance and the thoroughness of the empirical evaluation.